# Etv6 activates *vegfa* expression through positive and negative transcriptional regulatory networks in *Xenopus* embryos

Lei Li [1], Rossella Rispoli[1,2], Roger Patient[1], Aldo Ciau-Uitz [1] & Catherine Porcher [1]

VEGFA signaling controls physiological and pathological angiogenesis and hematopoiesis. Although many context-dependent signaling pathways downstream of VEGFA have been uncovered, *vegfa* transcriptional regulation in vivo remains unclear. Here, we show that the ETS transcription factor, Etv6, positively regulates *vegfa* expression during *Xenopus* blood stem cell development through multiple transcriptional inputs. In agreement with its established repressive functions, Etv6 directly inhibits expression of the repressor *foxo3*, to prevent Foxo3 from binding to and repressing the *vegfa* promoter. Etv6 also directly activates expression of the activator *klf4;* reflecting a genome-wide paucity in ETS-binding motifs in Etv6 genomic targets, Klf4 then recruits Etv6 to the *vegfa* promoter to activate its expression. These two mechanisms (double negative gate and feed-forward loop) are classic features of gene regulatory networks specifying cell fates. Thus, Etv6's dual function, as a transcriptional repressor and activator, controls a major signaling pathway involved in endothelial and blood development in vivo.

---

[1] MRC Molecular Haematology Unit, MRC Weatherall Institute of Molecular Medicine, Radcliffe Department of Medicine, University of Oxford, Oxford OX3 9DS, UK. [2] Present address: Division of Genetics and Molecular Medicine, NIHR Biomedical Research Centre, Guy's and St Thomas' NHS Foundation Trust and King's College London, London SE1 9RT, UK. Correspondence and requests for materials should be addressed to R.P. (email: roger.patient@imm.ox.ac.uk) or to A.C.-U. (email: aldo.ciau@imm.ox.ac.uk) or to C.P. (email: catherine.porcher@imm.ox.ac.uk)

Vascular endothelial growth factor A (VEGFA) signaling is critical for both physiological and pathological processes in the adult, including hematopoiesis, angiogenesis and solid tumour progression. In hematopoiesis, VEGFA regulates hematopoietic stem cell (HSC) survival and function in the bone marrow through both cell intrinsic and extrinsic mechanisms[1,2]. In angiogenesis, VEGFA is required for endothelial cell proliferation, migration and organization in three dimensions to form new vessels during vascular remodelling[3]. Finally, in cancer, VEGFA signaling plays essential roles in the regulation of tumour angiogenesis and metastasis[4], and has emerged as a key anti-angiogenic target in a wide variety of cancer therapies[5]. However, inhibitors of VEGFA-mediated signaling often trigger limited responses in both human tumours and mouse models of cancer, due to the development of evasive and intrinsic resistance mechanisms[5].

VEGFA signaling is also essential during embryogenesis where it plays pivotal roles in the development of the endothelial and hematopoietic systems. Knockout of *VegfA* or *VegfA* receptor (*Vegfr1*, *2*) genes in mice results in early lethality owing to severe defects in vascular development[6]. Mice deficient in *Vegfr2* (also known as *Flk1* or *Kdr*) show an absence of yolk sac blood islands and reduction of CD34$^+$ hematopoietic progenitors[7]. In vitro, VEGFA is required for the specification of FLK1$^+$ mesoderm into hematopoietic and cardiovascular lineages in mouse embryonic stem cell differentiation models[8,9]. In embryos, *vegfa* is essential for HSCs emergence and has been shown to be required at several stages of their programming[10–12]. In *Xenopus*, acting in a paracrine manner from the somites, Vegfa is required for the establishment of definitive hemangioblasts (DHs, precursors of the dorsal aorta (DA) and HSCs) in the dorsal lateral plate (DLP) mesoderm[11,12]. Later, Vegfa secreted by the hypochord guides the migration of DA precursors from the DLP to the midline[11,12].

Although critical to understand the development of the vascular and hematopoietic systems, very little is known about the regulation of the complex spatio-temporal expression pattern of *vegfa* during embryogenesis. Because developmental and tumour biology and, in particular, angiogenic and metastatic processes share signaling and transcriptional pathways[13], a better understanding of the regulatory networks lying upstream of VEGFA during specification of the hematovascular lineage may help identify new druggable targets relevant to VEGFA-dependent diseases.

ETV6, a transcriptional repressor belonging to the ETS family of transcription factors (TFs)[14–16], is a key regulator of hematopoietic, angiogenic and tumorigenic processes, and has been linked to *vegfa* regulation. It is essential for bone marrow hematopoiesis[17,18] and is involved in many chromosomal rearrangements which lead to childhood leukaemias[19]. Additionally, germline and somatic mutations in *ETV6* have recently been shown to be involved in other malignancies such as skin, colon, mammary and salivary gland cancers[20–22]. In mouse embryos, *Etv6* homozygous deletion leads to early embryonic death because of defective yolk sac angiogenesis[23].

Previously, we showed that Etv6 specifies DHs and HSCs in developing *Xenopus* embryos through positive regulation of *vegfa* expression in the somites at stage 22 of development[12]. Because Etv6 is considered a transcriptional repressor[14,15,24], this suggested an indirect regulatory mechanism of action whereby Etv6 directly represses expression of a *vegfa* transcriptional repressor. To test this hypothesis, we set out to investigate the mechanism by which Etv6 activates *vegfa* expression in the somites. Using genome-wide and functional analyses, we identify Etv6 direct target genes and show that Etv6 employs multiple regulatory mechanisms. To allow *vegfa* expression, Etv6 directly represses *foxo3*, a known transcriptional repressor of VegfA in cancer cells[25,26], thus participating in a double negative gate. Unexpectedly, Etv6 also directly activates expression of *klf4*, a known activator of VegfA in endothelial cells and a regulator of metastatic processes[27,28]. Consistent with a feed-forward loop mechanism, Etv6 also binds the *vegfa* promoter through a Klf4-dependent mechanism. Thus, our work demonstrates that *vegfa* expression in the somites is regulated by a complex gene regulatory network (GRN) with Etv6 acting both as a repressor and, unexpectedly, an activator. These findings further our understanding of the in vivo regulation of *vegfa* and provide a platform for the study of the processes underlying VEGFA-dependent tumorigenesis, such as those involving FOXO3 and KLF4, and for the establishment of future therapeutic strategies.

## Results

**Etv6 genome-wide occupancy in the somites of *Xenopus* embryos.** To investigate the mechanism by which Etv6 regulates *vegfa* expression in the somites, we first set out to identify Etv6 genomic targets through ChIP-seq (Fig. 1a). To this end, we generated ChIP-grade anti-*Xenopus*-Etv6 polyclonal antibodies in rabbits. Three short peptides were designed (Supplementary Figure 1a) and used to generate six polyclonal antibodies (two/peptide). Four of these antibodies detected Etv6 protein on western blots (Supplementary Figure 1b) and were tested for their ability to immunoprecipitate endogenous Etv6 protein from protein extracts generated from the somites of stage 22 embryos. Two antibodies (Etv6-2a and Etv6-2b) immunoprecipitated endogenous Etv6 protein (Supplementary Figure 1c). Etv6-2a showed greater affinity to Etv6 and, therefore, was selected for ChIP-seq experiments. Chromatin immunoprecipitation was performed on three independent biological samples obtained from the somites of wild-type (WT) stage 22 embryos (Fig. 1a). Indexed libraries were generated from immunoprecipitated DNA and control input, samples were pooled and sequenced, and reads mapped to the *X. laevis* genome.

A total of 9128 Etv6 consistent peaks across the three biological replicates were identified using DiffBind[29] (Supplementary Data 1). The genomic distribution of the peaks revealed Etv6 occupancy over distinct features including transcription start sites (TSS), transcription termination sites (TTS), exons, introns and intergenic regions (Fig. 1b). Additionally, 22.6% of the peaks mapped to genomic regions not yet annotated (NA). For all subsequent analyses, and as a first step to identify Etv6 direct transcriptional targets, we focused on the 2440 peaks (26.7%) located in TSS regions that could confidently be associated with 2416 genes. In Fig. 1c, to illustrate the quality of the Etv6 ChIP-seq data, we show the profiles of two genes with the highest enrichment of Etv6 in their TSS region (*rimklb* and *Xelaev18005411m*). *Tal1*, a hematopoietic gene not expressed in the somites at stage 22[11], is an example of a gene devoid of Etv6 peaks at its TSS region.

Next, we performed de novo motif analysis on the sequences underlying all Etv6 peaks associated with TSS regions using Homer software[30]. Etv6 peaks showed significant enrichment of consensus binding sequences corresponding to known TFs (Fig. 1d). Surprisingly, ETS family binding motifs were not represented (Supplementary Table 1). Instead, the most overrepresented motif belonged to the Klf/Sp family, followed by consensus motifs for Nfy, Creb, Meis and Fox TFs. A similar result was obtained when the de novo motif analysis was performed on all 9128 Etv6 peaks across the genome, as the most overrepresented binding motif corresponds to the Klf/Sp family (28.29%, Supplementary Data 2). Only a small fraction of the peaks (3.31%) was found to be enriched with ETS-binding motifs (TACTTCTT, Supplementary Data 2). This suggested that,

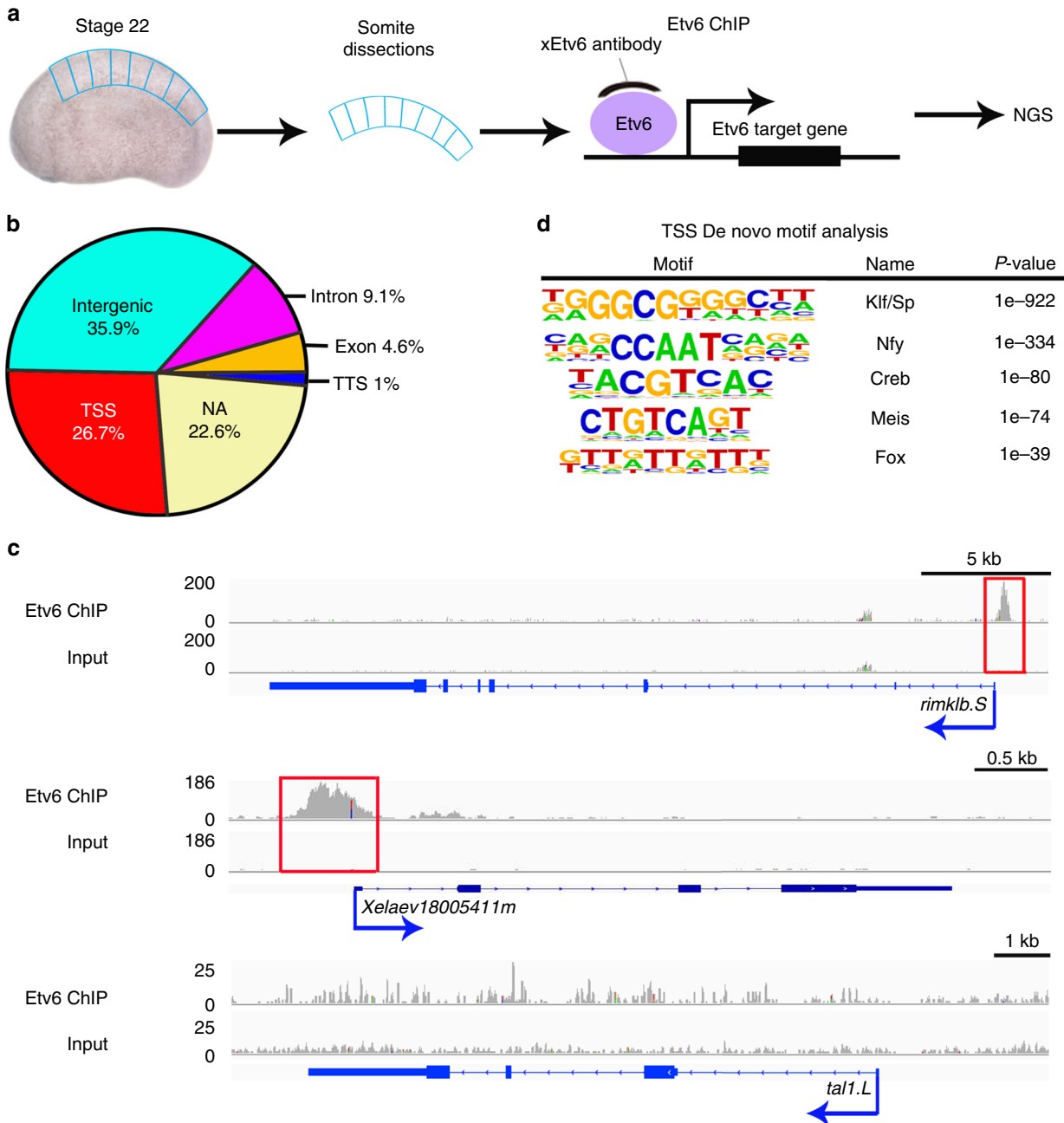

**Fig. 1** Genome-wide occupancy of Etv6 in *Xenopus* somites at stage 22. **a** Experimental design of Etv6 ChIP-seq assay. The somites were dissected from stage 22 *Xenopus laevis* embryos, homogenized and fixed, and subjected to Etv6 ChIP-seq. NGS, next-generation sequencing. **b** Genomic distribution of the 9128 Etv6 ChIP-seq peaks throughout the *X. laevis* genome. NA, not annotated; TSS, transcription start site; TTS, transcription termination site. **c** Integrative genome viewer (IGV) showing examples of Etv6 ChIP-seq tracks. Etv6 binding is enriched in the TSS region of *rimklb.S* and *Xelaev18005411m* (red boxes) but not in the TSS region of *tal1.L*. Input is shown as control. The consensus peaks were generated using DiffBind; for the peak tracks of individual replicates, see Supplementary Figure 13. **d** De novo motif analysis of ETV6 peaks located in the TSS region. Five out of the 22 overrepresented motifs are shown

globally, Etv6 is preferentially recruited to chromatin indirectly, through interaction with other DNA-binding TFs such as members of the Klf/Sp family.

**Establishing the somitic transcriptome regulated by Etv6.** In *Xenopus* embryos, *vegfa* expression in the somites at stage 22 of development is completely dependent on Etv6[12]. To determine the transcriptional targets of Etv6 in this tissue and stage of development, we compared the transcriptome of WT and Etv6-deficient somites, the latter generated with a previously validated

*etv6* antisense morpholino oligonucleotide (MO)[12] (Fig. 2a). Spearman correlation analysis and principal component analysis (PCA) revealed that *etv6* deficiency causes significant changes in the transcriptome of the somites (Fig. 2b, Supplementary Figure 2). Indeed, differential expression analysis identified a total of 5186 differentially expressed genes (DEGs, FDR <0.05) with 2236 genes normally repressed by Etv6 (upregulated in Etv6-deficient somites) and 2950 genes normally activated by Etv6 (down-regulated in Etv6-deficient somites) (Fig. 2c, Supplementary Data 3). Interestingly, gene ontology analysis indicated an enrichment in categories related to positive and negative

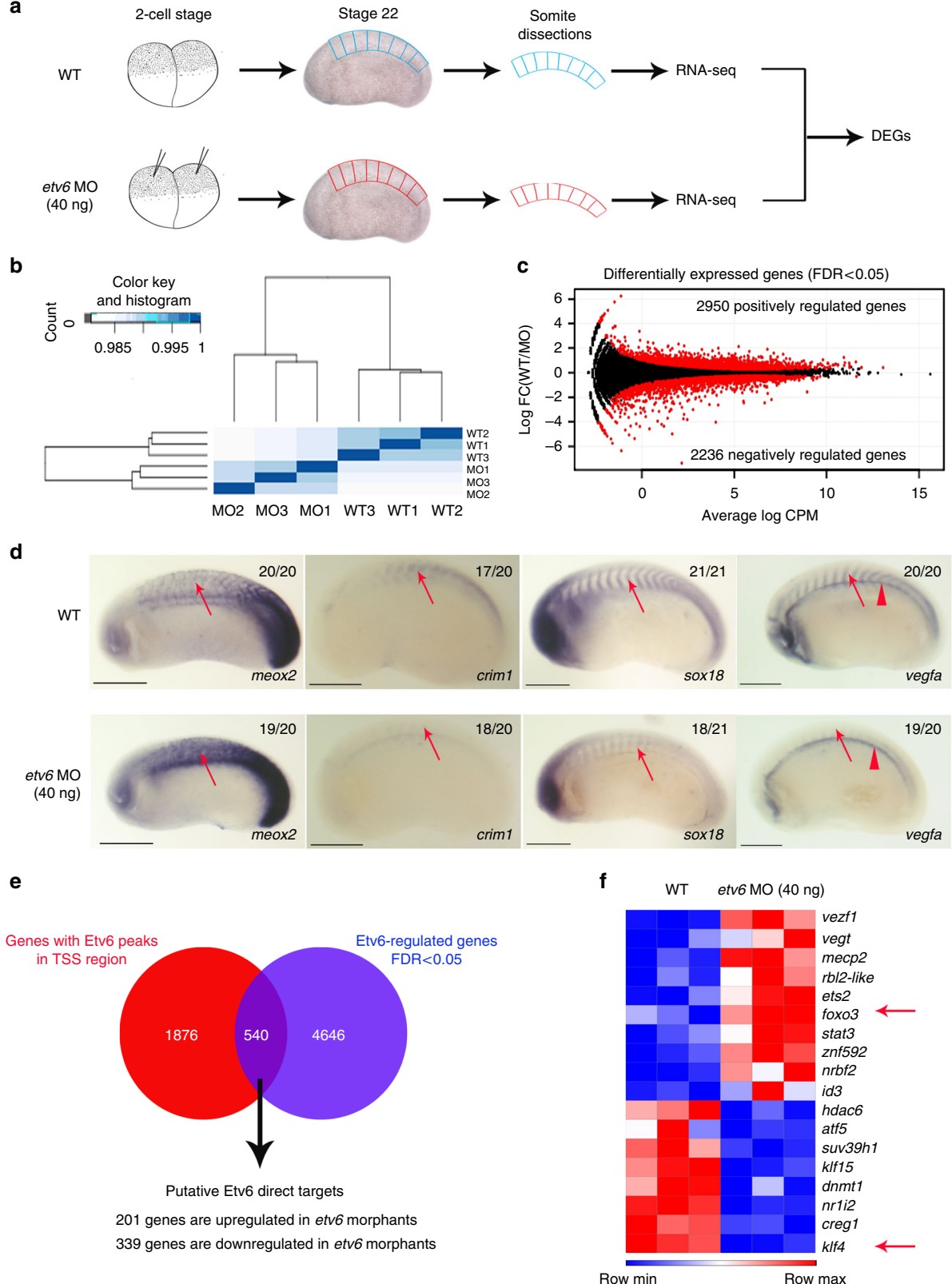

regulation of transcription (Supplementary Data 4), confirming our hypothesis that Etv6 controls biological processes through the regulation of the expression of transcriptional regulators.

To validate the RNA-seq data, we next confirmed the differential expression of selected genes in WT and Etv6-deficient embryos by Whole-mount in situ hybridization (WISH)

(Fig. 2d). During vertebrate development, the homeobox genes *meox1* and *meox2* regulate somitogenesis and myogenesis[31]. In zebrafish, *meox1* also represses the expansion of somite-derived endothelial cells thereby limiting HSC development in the DA[32]. The single *Xenopus meox* gene is repressed by Etv6 in somites (WT/*etv6* MO log$_2$FC = −0.89 for *meox2.L* and −0.53 for

**Fig. 2** Identification of Etv6 direct transcriptional targets in the somites. **a** Experimental design. RNA-seq was performed on wild type (WT) and Etv6-deficient (*etv6* MO-injected embryos) somite explants dissected from stage 22 embryos. Differentially expressed genes (DEGs) were identified by comparing the transcriptome of these tissues. **b** Spearman correlation analysis on triple biological RNA-seq replicates. **c** MA plot (magnitude of the difference versus amplitude of the signal) showing the fold-change in gene expression of all genes in WT versus Etv6-deficient somites ($log_2FC$) compared to their expression levels ($log_2CPM$). Black dots, non-significant change; red dots, differentially expressed genes (DEGs). The number of positively and negatively regulated DEGs is indicated. **d** WISH showing the expression of DEGs in stage 22 WT and Etv6-deficient embryos. *Meox2* expression is upregulated in the somites (arrows) in Etv6-deficient embryos whereas expression of *crim1*, *sox18* and *vegfa* is downregulated. Note that *vegfa* expression in the hypochord is unaffected (arrowheads). Embryos are shown in lateral view with anterior to the left and dorsal to the top. Numbers in top right corner indicate the number of embryos exhibiting the phenotype pictured (scale bars: 0.5 mm). **e** The intersection between DEGs and genes harbouring Etv6 ChIP-seq peaks in their TSS region reveals 540 putative direct target genes. **f** *Foxo3* and *klf4* (arrows), known transcriptional regulators of *vegfa*, are amongst the 18 TFs and chromatin modifiers identified as potential direct transcriptional targets of Etv6

*meox2.S*) and WISH confirmed higher expression in Etv6-deficient somites (Fig. 2d). *Crim1*, a regulator of VEGFA autocrine signaling in endothelial cells[33], is positively regulated by Etv6 (WT/*etv6* MO $log_2FC = 0.68$). WISH analysis confirmed downregulation of *crim1* expression in the somites of Etv6-deficient embryos (Fig. 2d). *Sox18*, a TF required for the development of blood vessels[34], is positively regulated by Etv6 in the somites, as revealed by RNA-seq and confirmed by WISH (WT/*etv6* MO $log_2FC = 1.01$, and decreased expression in the somites of Etv6-deficient embryos, Fig. 2d). Finally, we have previously reported that *vegfa* expression in the somites is activated by Etv6[12]. Intriguingly, *vegfa* was not identified as a differentially expressed gene by RNA-seq. WISH analysis did, however, confirm that *vegfa* is absent in the somites of Etv6-deficient embryos. It also revealed strong *vegfa* expression in the hypochord of both WT and Etv6-deficient embryos (Fig. 2d), thus providing an explanation for the absence of *vegfa* from the DEG list, as the explants used for RNA extraction contained the hypochord, the unperturbed expression of *vegfa* in the Etv6-deficient hypochord masked its downregulation in the somites. More importantly, this demonstrates that Etv6 specifically controls the expression of *vegfa* in the somites. In conclusion, our RNA-seq data reveals a robust transcriptional response to *etv6* knock-down.

**Identification of Etv6 direct target genes.** To identify Etv6 direct transcriptional targets, we compared the list of genes harbouring Etv6 peaks in their TSS regions (2416 genes) to the list of DEGs (5186 genes). This revealed 540 putative direct target genes with 545 peaks in their TSS regions (Fig. 2e, Supplementary Data 5). Surprisingly, around two thirds of Etv6 target genes (339/540, 63%) were downregulated in the somites of Etv6-deficient embryos, strongly indicating that Etv6, typically considered as a transcriptional repressor[19], can also act as an activator of gene expression in the somites at stage 22. De novo motif analysis on the sequences underlying the peaks linked to genes normally activated by Etv6 identified Klf/Sp, Nfy and Creb motifs. Notably, only 9.94% of these peaks contained ETS motifs (CTTCCGCCCTTT and GGCCGGCAGTGT, Supplementary Table 2). Regarding the genes normally repressed by Etv6, their Etv6 peaks contained Klf/Sp, Nfy, Creb and Hox motifs, and were not enriched in ETS-binding *cis*-elements (Supplementary Table 2). This is similar to what was observed in de novo motif analysis on all Etv6 ChIP-seq peaks and peaks at TSSs: ETS-binding motifs are present in only a small fraction of the peaks (see above and Supplementary Table 1 and Supplementary Data 2). Therefore, Etv6 recruitment to all its genomic targets, including the genes it normally regulates transcriptionally, largely depends on additional DNA-binding TFs.

To identify the transcriptional regulators that could mediate Etv6 control of *vegfa* expression, we next focused on the 18 TFs and chromatin modifiers detected amongst the Etv6 putative direct targets with fold-change in gene expression >1.5 (Fig. 2f).

Many of these transcriptional regulators are involved in the control of *vegfa* expression in higher vertebrates[35–39]. Amongst those, *foxo3* (Forkhead box o3), a transcriptional repressor of *vegfa* in cancer cells[25,26], and *klf4* (Krüppel-like factor 4), a transcriptional activator of *vegfa* in endothelial cells and a regulator of metastatic processes[27,28], are known to exert their functions through direct binding to the *VegfA* promoter[25,27]. In *Xenopus* somites, their expression is respectively repressed and activated by Etv6 (Fig. 2f). For these reasons, we hypothesized that Foxo3 and Klf4 may directly contribute to the regulation of *vegfa* expression in the somites downstream of Etv6.

**Etv6 prevents Foxo3-mediated repression of *vegfa*.** To investigate whether Etv6 could be regulating *vegfa* expression by repressing *foxo3* expression, we first confirmed the binding of Etv6 to both *foxo3* genes in the *X. laevis* genome. *X. laevis* is an allotetraploid organism with a genome organized into two homologous sub-genomes, referred to as L and S, and most genes (>56%) are represented by two distinct homologous copies which can be regulated differently[40]. ChIP-seq tracks in Fig. 3a show that Etv6 peaks are present in the TSS region of both *foxo3.L* and *foxo3.S* and Etv6 binding in these regions was confirmed by ChIP-qPCR (Fig. 3b). These peaks contained one Ets (5'-G(A/T)GGAAG(G/T)-3') and several Klf/Sp binding motifs. Furthermore, upregulation of *foxo3* in the somites of Etv6-deficient embryos was confirmed by WISH with probes designed to target both *foxo3.L* and *foxo3.S* (Fig. 3c) and RT-qPCR on RNA extracted from stage 22 somites with primers designed to detect both *foxo3* genes (Fig. 3d). To functionally link the repressive function of Etv6 on *foxo3* expression to Etv6-occupied DNA regions, we next tested the transcriptional activity of the sequences underlying the Etv6 peaks in *foxo3.L* and *foxo3.S* promoters in luciferase assays. When compared to the empty vector (control-*luc*), both sequences had the capacity to induce luciferase activity in wild type embryos, with the *foxo3.S* sequence exhibiting a stronger transcriptional activity (Supplementary Figure 3). Critically, luciferase activity was significantly enhanced in Etv6-deficient embryos, confirming that these sequences are involved in Etv6-mediated repression of *foxo3*. In conclusion, both *foxo3.L* and *foxo3.S* are directly repressed by Etv6 in the somites.

Next, we tested the hypothesis that Foxo3 may repress the expression of *vegfa* in the somites. First, we determined whether upregulation of *foxo3* was sufficient to repress *vegfa* in the somites by injecting *foxo3* mRNA into 2-cell stage embryos and testing *vegfa* expression by WISH at stage 22. Indeed, *vegfa* expression was dramatically downregulated in the somites of embryos injected with exogenous *foxo3* mRNA (Fig. 3e) whereas expression in the hypochord was unaffected. Importantly, *etv6* expression was unaffected in the somites of *foxo3*-overexpressed embryos (Supplementary Figure 4). This indicated that *foxo3*, like Etv6, specifically regulates the expression of *vegfa* in the somites,

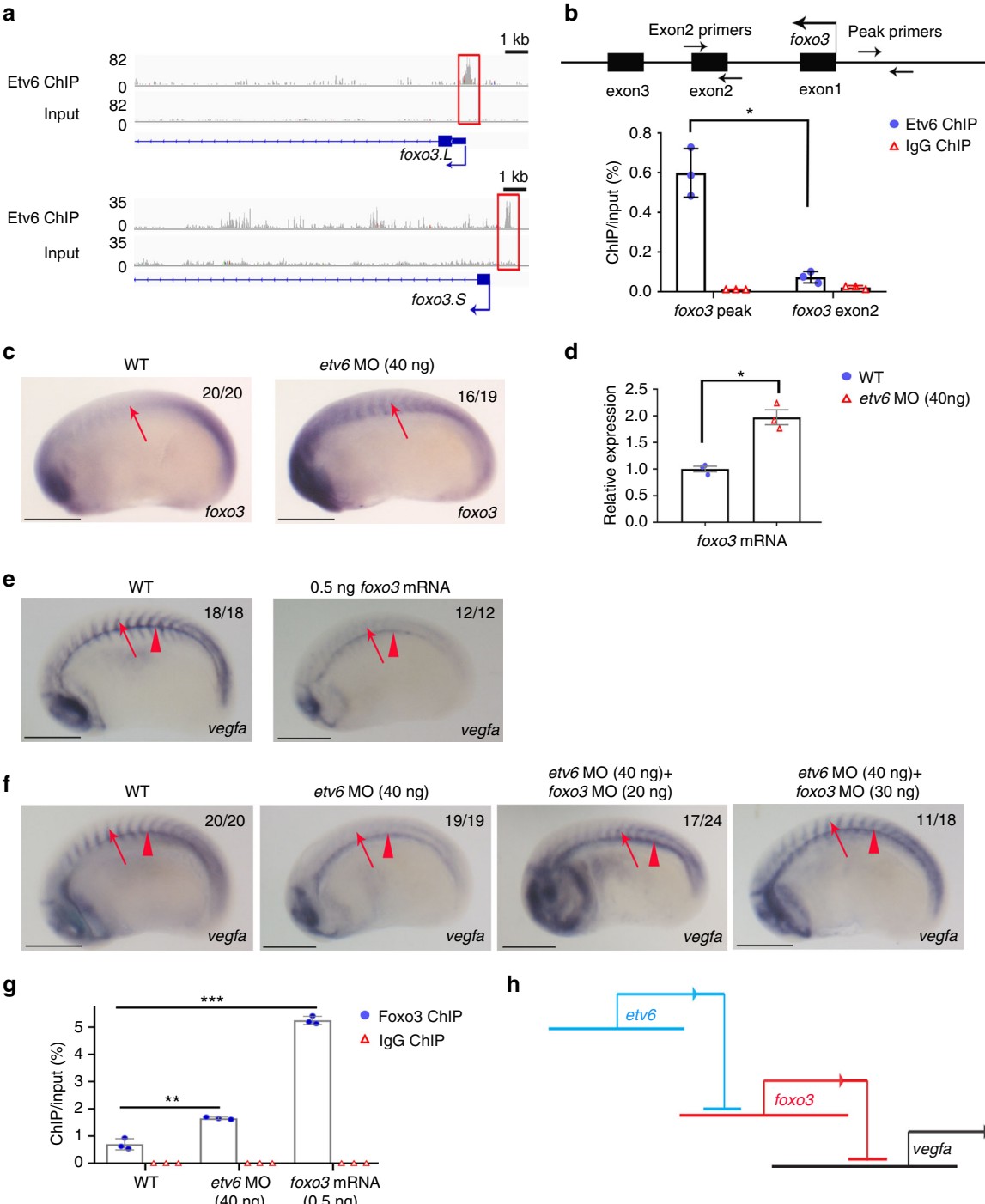

supporting the notion that Etv6 positively regulates the expression of *vegfa* through repression of *foxo3*. To further confirm that *vegfa* downregulation in the somites of Etv6-deficient embryos is mediated by Foxo3, we designed MOs that efficiently block the translation of *foxo3* (Supplementary Figure 5a–b). Consistent with the absence of *foxo3* expression in the somites of stage 22 embryos (Fig. 3c and Supplementary Figure 6), injection of *foxo3* MOs had no effect on *vegfa* expression in the somites of WT embryos (Supplementary Figure 5c). However, co-injection of *foxo3* MOs with *etv6* MOs rescued the expression of *vegfa* in the somites of Etv6-deficient embryos (Fig. 3f), indicating that Etv6 prevents Foxo3-mediated repression of *vegfa* in the somites.

In cancer cells, FOXO3 has been reported to repress *VegfA* expression by directly binding to the *VegfA* promoter region[25,26]. Using Jaspar[41], we have identified a putative Foxo3 binding site (5'-GTAAACA-3')[42] in the promoter region of *Xenopus vegfa* distantly localized upstream the Etv6 ChIP peak (Supplementary Data 6). Foxo3 ChIP-qPCR experiments showed that Foxo3 does indeed bind this sequence and, as expected, occupancy is significantly increased in Etv6-deficient embryos or when *foxo3* is overexpressed in embryos (Fig. 3g). Furthermore, in luciferase assays, a *vegfa* promoter fragment containing the Foxo3 binding motif can indeed repress transcriptional output and, critically, its repressive activity is lost when the Foxo3 binding motif is deleted (Supplementary Figure 7). Taken together, these results indicate that *foxo3* directly represses the transcriptional activity of the *vegfa* promoter (Fig. 3h).

**Fig. 3** Etv6 prevents Foxo3-mediated repression of *vegfa* in the somites. **a** IGV showing Etv6 peaks (red boxes) in the TSS region of *foxo3.L* and *foxo3.S*. Input is shown as control. For the peak tracks of individual replicates, see Supplementary Figure 13. **b** ChIP-qPCR analysis confirming that Etv6 binding is enriched in the *foxo3* promoter region. The diagram above the histogram depicts the *foxo3* locus and the location of the primers used to amplify the region of the Etv6 peak in the *foxo3* TSS region (peak primers) and a negative control region in exon2 (exon2 primers). Primers were designed to target both *foxo3.L* and *foxo3.S*. IgG ChIP was used as negative control. Error bars represent SEM of three biological replicates. *$P = 0.014$, two-tailed Student's *t*-test. **c** WISH showing that *foxo3* expression is upregulated in the somites (arrows) of Etv6-deficient (*etv6* MO) embryos (scale bars: 0.5 mm). **d** RT-qPCR confirming that *foxo3* is upregulated in stage 22 Etv6-deficient somites. Expression was normalized to *odc1*. Error bars represent SEM of three biological replicates. *$P = 0.020$, two-tailed Student's test. **e** WISH showing that overexpression of *foxo3* (0.25 ng *foxo3.L* mRNA + 0.25 ng *foxo3.S* mRNA) blocks *vegfa* expression in the somites (arrows) whereas expression in the hypochord (arrowheads) is unaffected (scale bars: 0.5 mm). **f** WISH showing that blocking *foxo3* translation with MOs rescues *vegfa* expression in the somites (arrows) of Etv6-deficient (*etv6* MO) embryos. Arrowheads indicate expression in the hypochord (scale bars: 0.5 mm). **g** Foxo3 ChIP-qPCR on the *vegfa* promoter region. The assay was performed on somitic material isolated from wild-type, Etv6-deficient (*etv6* MO) and *foxo3* mRNA overexpressing embryos at stage 22. IgG ChIP was used as negative control. Error bars represent SEM of three biological replicates. **$P = 0.0014$, ***$P = 0.00018$, two-tailed Student's *t*-test. **h** Diagram illustrating that *vegfa* expression in the somites requires the repression of *foxo3* by Etv6, i.e., Etv6 represses a repressor of *vegfa*. Images in **c**, **e**, **f** show stage 22 embryos in lateral view with anterior to the left and dorsal to the top. Numbers in top right corner indicate the number of embryos exhibiting the phenotype pictured

**Etv6 acts as a direct transcriptional activator of *klf4*.** The *klf4* gene is a candidate direct target of Etv6 in the somites (Fig. 2f). Interestingly, KLF4 has been shown in human retinal microvascular endothelial cells (HRMECs)[27] and human umbilical vein endothelial cells (HUVECs)[28] to activate *VegfA* expression through direct binding to its promoter. We, therefore, hypothesized that Etv6 could activate *vegfa* expression in the somites through Klf4.

We first validated binding of Etv6 in the promoter regions of the two *Xenopus klf4* genes, *klf4.L* and *klf4.S* (Fig. 4a, b), and confirmed by WISH (Fig. 4c) and RT-qPCR (Fig. 4d) on stage 22 Etv6-deficient embryos that expression of *klf4* in the somites was dependent on Etv6. Additionally, we demonstrated by western blot that Klf4 protein levels were dramatically reduced in the somites of Etv6-deficient embryos (Fig. 4e, *etv6* MO). Finally, to further confirm that the transcriptional activity of the *klf4* promoter is regulated by Etv6 in the somites, the DNA fragment corresponding to the Etv6 peak in the *klf4* promoter was functionally tested in in vivo luciferase assays in both wild-type and Etv6-deficient backgrounds. We observed a significant decrease of the transcriptional activity of this sequence in Etv6-deficient embryos (Supplementary Figure 3). Taken together, these data indicate that *klf4* expression in the somites is directly activated by Etv6.

Analysis of expression of *klf4* during early embryogenesis shows a remarkably transient expression in the somites where it is only detected at stage 22 (Supplementary Figure 8). Importantly, *vegfa* expression in the somites is absolutely dependent on Etv6 at this stage[11]. This is also the stage when *vegfa* secreted from the somites is required for the programming of definitive hemangioblasts in the lateral plate mesoderm[12]. In order to investigate the role of Klf4 in *vegfa* expression and HSC programming, we designed a MO which efficiently blocks the in vivo translation of Klf4 (Fig. 4e, *Klf4* MO) as well as TALENs (transcription activator-like effector nucleases) which efficiently generate null mutations in the second exon of *klf4* (Fig. 4f). Klf4-deficient embryos generated either by MO or TALEN injection showed a dramatic downregulation of *vegfa* expression in the somites at stage 22 (Fig. 4g). Thus, Klf4 is required for *vegfa* expression in the somites. Furthermore, definitive hemangioblast specification in the lateral plate mesoderm, as indicated by *tal1* expression at stage 26–28, and hemogenic endothelium emergence in the dorsal aorta, as indicated by *runx1* expression at stage 39, failed in Klf4-deficient embryos (Supplementary Figure 9). Therefore, Klf4 is essential for *vegfa* expression in the somites and the programming of HSCs during embryogenesis, thus recapitulating the Etv6 functions.

Collectively, our studies show that Etv6 positively regulates the expression of *vegfa* in the somites through direct transcriptional activation of *klf4*, a transcriptional activator of *vegfa* (Fig. 4h).

**Klf4 is required for Etv6 binding to the *vegfa* promoter.** Analysis of Etv6 ChIP-seq peaks associated with the *vegfa* genes revealed binding of Etv6 in the *vegfa.L* (Fig. 5a), but not *vegfa.S* (Supplementary Figure 13), promoter region (−552 to −244 bp upstream of *vegfa.L* TSS). Analysis of the genomic structure of *vegfa.S* indicated that it is a gene undergoing degeneration and it is not transcribed during embryogenesis (Xenbase expression data after Sessions et al.[40]). Thus, only one *vegfa* gene, *vegfa.L*, is functional in *X. laevis* (referred to as *vegfa* hereafter) and bound by Etv6 at its TSS. To test whether *vegfa* could be directly bound by Etv6, we performed Etv6 ChIP-qPCR on the *vegfa* promoter and confirmed the enrichment of Etv6 in this region (Fig. 5b, c). However, we did not find ETS-binding motif under this peak, suggesting that Etv6 cannot directly bind to the *vegfa* promoter. Instead, two binding motifs for the Klf/Sp family with high matrix scores were identified (Supplementary Data 6). This is consistent with our de novo motif analysis showing that Klf/Sp binding motifs are the most overrepresented sequences.

As Klf4 is required for *vegfa* expression in the somites, we wondered whether it was involved in Etv6 binding to the *vegfa* promoter. Therefore, we performed Klf4 ChIP-qPCR on the *vegfa* promoter using the same primers as for Etv6 ChIP-qPCR (Fig. 5b, c). This confirmed that Klf4 binding is indeed enriched in the promoter of *vegfa* and co-localises with Etv6 binding (Fig. 5d), supporting the hypothesis that Klf4 is involved in Etv6 recruitment to the *vegfa* promoter. To further confirm this notion, Etv6 ChIP-qPCR was carried out on Klf4-deficient somites (Fig. 5e). Etv6 enrichment in the *vegfa* promoter was severely reduced in Klf4-deficient embryos, a reduction similar to that observed in Etv6-deficient embryos (Fig. 5e). Importantly, *etv6* expression was unaffected in the somites of Klf4-depleted embryos (Supplementary Figure 10). Therefore, Klf4 is required for Etv6 recruitment to the *vegfa* promoter.

To functionally test the role of the two Klf4 binding motifs identified under the Etv6 peak in the *vegfa* promoter, we designed TALENs that efficiently deleted them (Fig. 5f, Supplementary Figure 11). In the somites of stage 22 embryos with Klf4 binding motifs deleted, the enrichment of Klf4 and Etv6 in the *vegfa* promoter region was dramatically decreased (Fig. 5g). Importantly, *vegfa* expression in the somites was downregulated in embryos deleted for Klf4 binding motifs, whereas expression in the hypochord was not affected (Fig. 5h), strongly indicating that these Klf4 binding sites are specifically required for *vegfa* expression in the somites of stage 22 embryos.

We have shown that Klf4 is required for the recruitment of Etv6 to the *vegfa* promoter (Fig. 5e). To test whether this occurs through direct physical interaction between these TFs, we performed co-immunoprecipitation (co-IP) assays (Fig. 6). First,

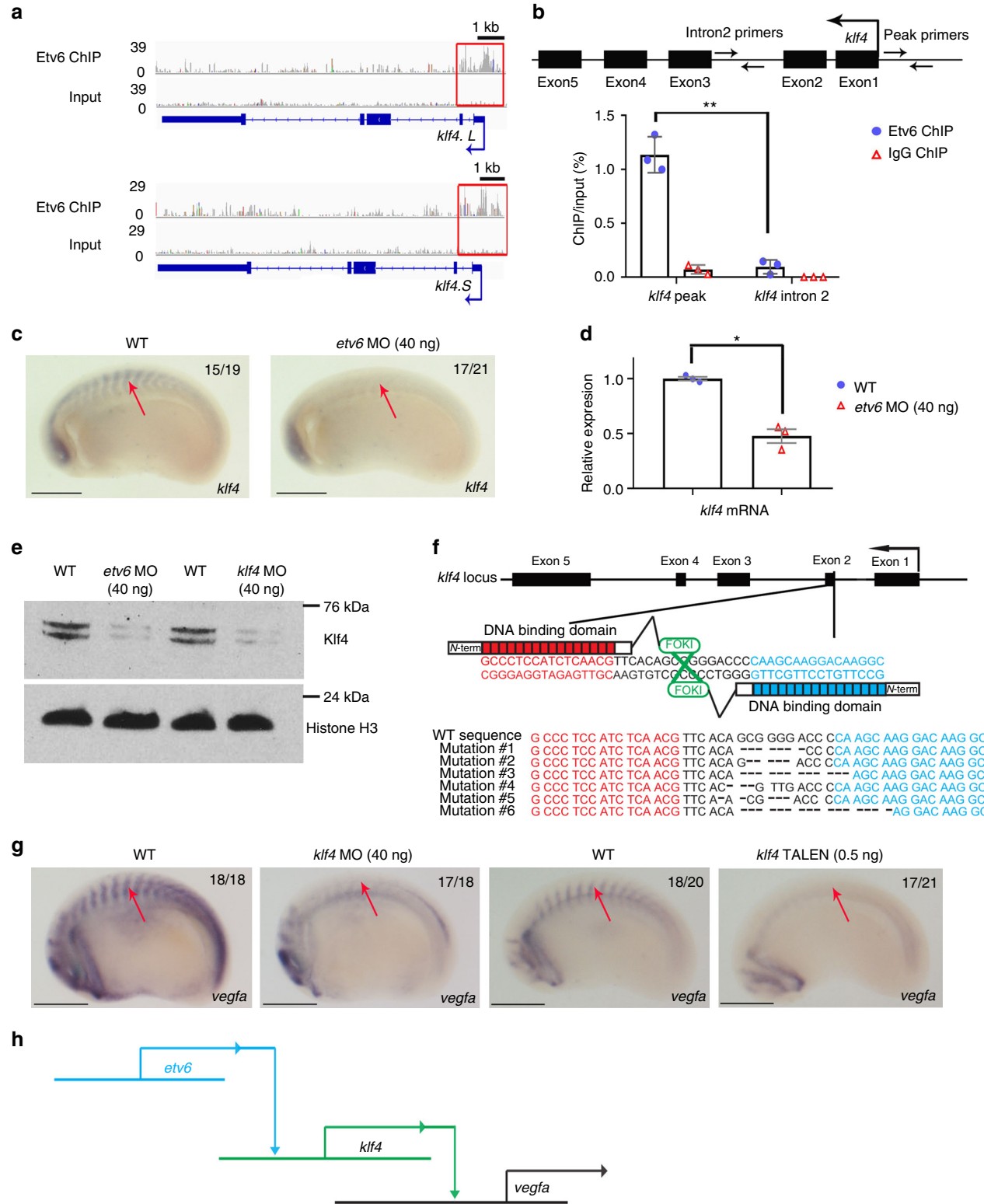

Etv6 was overexpressed in developing embryos and its ability to co-immunoprecipitate endogenous Klf4 was assessed. Western blot analysis of Etv6 immunoprecipitates showed co-purification of Klf4 (Fig. 6a). To ask whether the interaction between Etv6 and Klf4 could be direct, mRNAs encoding these proteins were translated in vitro using the reticulocyte lysate system and the translated proteins were used directly in co-IP assays. In vitro translated Etv6 and Klf4 did not co-immunoprecipitate

(Fig. 6b), in contrast to two TFs known to directly interact, Lmo2 and Ldb1[43], used as positive control (Fig. 6c).

The expression of *vegfa* in the somites and the establishment of definitive hematopoiesis are impaired in both Etv6- and Klf4-deficient embryos. As expression of *etv6* in Klf4-deficient embryos is unaffected (Supplementary Figure 10), we hypothesized that Klf4 could be the main regulator of these processes. To test this, we attempted to rescue the haematopoietic phenotype of

**Fig. 4** Etv6 positively regulates *vegfa* expression through transcriptional activation of *klf4*. **a** IGV showing Etv6 peaks (red boxes) in the TSS region of *klf4.L* and *klf4.S*. Input is shown as control. For the peak tracks of individual replicates, see Supplementary Figure 13. **b** ChIP-qPCR analysis confirming that Etv6 is enriched in the *klf4* promoter region. The diagram above the histogram depicts the *klf4* locus and the location of the primers used to amplify the region of the Etv6 peak in the *klf4* TSS region (peak primers) and a negative control region in intron2 (intron2 primers). Primers were designed to target both *klf4.L* and *klf4.S*. IgG ChIP was used as negative control. Error bars represent SEM of three biological replicates. **P = 0.0036, two-tailed Student's t-test. (**c**) WISH showing that *klf4* expression in the somites (arrows) is downregulated in Etv6-deficient embryos (scale bars: 0.5 mm). **d** RT-qPCR confirming that *klf4* is downregulated in stage 22 Etv6-deficient somites. Expression was normalized to *odc1*. Error bars represent SEM of three biological replicates. *P = 0.020, two-tailed Student's t-test. **e** Western blot showing that Klf4 protein is depleted in Etv6-deficient embryos and that *klf4* MO blocks efficiently the translation of Klf4. Histone H3 was used as a loading control. **f** *Klf4* TALEN design. Top, diagram showing the sequences in *klf4* exon 2 targeted by the TALENs; bottom, DNA alignment showing the range of mutations generated by TALEN activity. Genomic DNA was obtained from stage 22 WT and TALEN-injected somites, and subjected to Sanger sequencing. TALEN-injection caused mutations in 78% of the clones sequenced. **g** WISH showing that *vegfa* expression in the somites (arrows) is downregulated in both *klf4* MO- and *klf4* TALEN-injected embryos (scale bars: 0.5 mm). **h** Diagram illustrating that Etv6 positively regulates *vegfa* expression in the somites through transcriptional activation of *klf4*, a transcriptional activator of *vegfa*

Etv6-deficient embryos by overexpressing *klf4* mRNA. Exogenous *klf4* was unable to rescue *vegfa, tal1* or *runx1* expression (Supplementary Figure 12), thus strongly supporting a model in which cooperation between Etv6 and Klf4 is required in vivo for the development of the HSC lineage.

Taken together, our results show that *klf4* is not only a direct target of Etv6, but also acts as its recruiting factor to the *vegfa* promoter to activate its expression.

## Discussion

In this study, we show that expression of the important signaling molecule *vegfa* is under tight transcriptional control by the ETS TF Etv6 during the early stages of hematovascular development. Previously, we showed that Etv6 is essential for expression of *vegfa* in the somites[12]. We now demonstrate that it works through both repressive and activating transcriptional mechanisms. As discussed below, this reveals unexpected molecular mechanisms engaged by Etv6 and a complex GRN upstream of *vegfa* in vivo.

To establish how Etv6 controls *vegfa* expression in the somites, we identified Etv6 direct target genes by focusing on genes that are both abnormally expressed in Etv6-deficient somites and bound by Etv6 in their TSS. This stringent approach revealed numerous biologically relevant targets. Indeed, many of the TFs and chromatin modifiers identified amongst Etv6's 540 direct target genes have previously been directly or indirectly implicated in the regulation of VegfA in endothelial and cancer cells. In addition to Foxo3 and Klf4 (the two targets examined in this study), these are Ets2, shown to down-regulate VegfA expression in HUVECs treated with arsenic trioxide (ATO)[39], the tumour suppressor Rbl2 (also known as RB2/p130), that inhibits tumour formation in mice by inhibiting VegfA-mediated angiogenesis[37], the transcriptional repressor Mecp2, that represses *VegfA* expression by binding to methylated CpG islands in the *VegfA* promoter[35], and the transcriptional regulator Creg1, that regulates human endothelial homoeostasis in vivo and promotes vasculogenesis in mouse ES cells through the activation of VegfA[36]. These findings validate our experimental design and the robustness of the list of Etv6's direct target genes.

ETV6 and its Drosophila orthologue, Yan, are well-established transcriptional repressors[14,15,24]. Surprisingly, in *Xenopus* somites, around two thirds (339/540, 63%) of Etv6 direct target genes are normally transcriptionally activated. Therefore, during the early stages of specification of the HSC lineage, Etv6 acts both as a transcriptional activator and a repressor. As detailed below, molecular and functional examination of Etv6 target genes revealed that Etv6's dual function is required for the regulation of *vegfa* expression.

Members of the ETS family of TFs are defined by a highly conserved ETS domain that recognizes a core sequence 5'-GGA

(A/T)-3' motif within the context of a 9- to 10-bp DNA sequence[44]. As all ETS TFs bind to the same motif, additional mechanisms regulating the selection of specific transcriptional targets within biological contexts are required. It has been suggested that high-affinity ETS motifs found in the promoter of housekeeping genes can be bound by any ETS TF whereas lower-affinity ETS-binding sites found in tissue-specific promoters and only bound by a subset of ETS TFs are flanked by binding sites of other TFs[45,46]. Cooperative binding with other TFs in sequences with composite binding sites results in a higher affinity and stable binding to DNA, and in synergistic repression or activation of specific target genes[47,48].

Here, de novo motif analysis of the 545 peaks associated with the promoters of genes normally activated or repressed by Etv6 showed (i) high prevalence of Klf/Sp binding sequences in both sets of targets; (ii) absence of ETS-binding motif enrichment in the peaks associated with genes repressed by Etv6; and (iii) ETS-binding motif enrichment in the peaks associated with a small fraction of genes activated by Etv6 (9.94%). This suggested that members of the Klf/Sp family are required for both the activating and repressive transcriptional activities of Etv6 and that Etv6 could be recruited to DNA through its interaction with other DNA-binding TFs. What determines Etv6 to act either as a transcriptional repressor or an activator remains to be fully explored.

Strikingly, *klf4* was one of the target genes activated by Etv6 in somites. KLF4 is a multifunctional TF which is critical for pluripotency, tissue homoeostasis, stemness, maintenance of cancer stem cells and normal hematopoiesis[49], and has previously been implicated in the expression of *VegfA* in human and mouse cells both positively and negatively. KLF4 binds the *VegfA* promoter and represses *Vegfa* expression during EMT in mouse mammary gland cells[50], whereas, in HRMECs and HUVECs, it promotes angiogenesis by transcriptionally activating *VegfA*[27,28]. Here, we show that Klf4 not only binds to conserved Klf/Sp motifs but also recruits Etv6 to the *vegfa* promoter region to activate *vegfa* expression. Our in vivo and in vitro co-immunoprecipitation (co-IP) assays indicate that these two proteins form part of the same transcriptional complex controlling gene expression in the *Xenopus* somites but are likely to interact indirectly. In support of this, it has been shown, in overexpression experiments with tagged proteins, that the ETS protein ERG co-immunoprecipitates with KLF2[51]. Similarly, in situ proximity ligation assays have demonstrated that KLF4 and the phosphorylated form of the ETS TF, ELK1, physically interact in human coronary artery cells[52]. Moreover, analysis of EHF (an ETS TF structurally related to ETV6) ChIP-seq peaks indicates that a significant number of peaks contained no ETS-binding motifs and many of them harboured Klf4/5 binding motifs only[53]. Interestingly, EHF is involved in the maintenance of

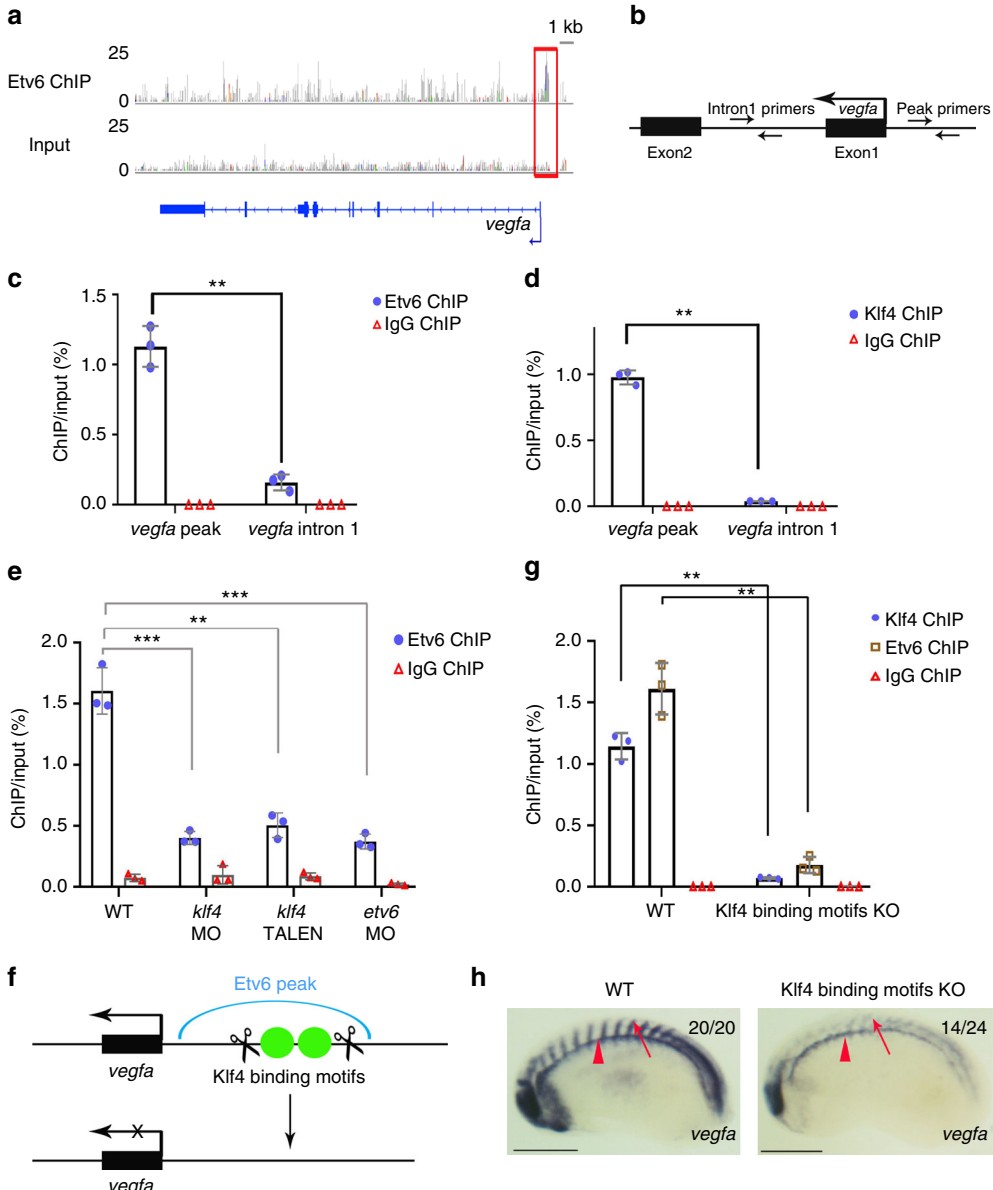

**Fig. 5** Klf4 is required for the recruitment of Etv6 to the *vegfa* promoter. **a** IGV view of Etv6 peaks reveals a peak (red box) in the TSS region of *vegfa*. Input is shown as control. For the peak tracks of individual replicates, see Supplementary Figure 13. **b** Partial representation of the *vegfa* locus showing the location of the primers used to amplify the region of the Etv6 peak in the *vegfa* TSS region (peak primers) and a negative control region in intron1 (intron1 primers). **c**, **d** ChIP-qPCR analysis on stage 22 somite explants confirming that Etv6 (**c**) and Klf4 (**d**) are enriched in the promoter region of *vegfa*. IgG ChIP was used as negative control. Error bars represent SEM of three biological replicates. **P = 0.003 (**c**), **P = 0.0011 (**d**), two-tailed Student's t-test. **e** The enrichment of Etv6 in the *vegfa* promoter is significantly reduced in Klf4-depleted (40 ng *klf4* MO and 0.5 ng *klf4* TALEN) stage 22 somites. This reduction is comparable to that produced by the depletion of Etv6 (40 ng *etv6* MO), strongly indicating that Klf4 is required for Etv6 recruitment to the *vegfa* promoter. IgG ChIP was used as negative control. Error bars represent SEM of three biological replicates. ***P = 0.0002 WT vs *klf4* MO, **P = 0.002 WT vs *klf4* TALEN, ***P = 0.000065 WT vs *etv6* MO, two-tailed Student's *t*-test. **f** Diagram showing the TALENs designed to delete the Klf4 binding motifs under the Etv6 peak in the *vegfa* promoter region. **g** The enrichment of Klf4 and Etv6 in the *vegfa* promoter region is significantly reduced in the somites of Klf4 binding motifs knockout (KO) embryos. IgG ChIP was used as negative control. Error bars represent SEM of three biological replicates. **P = 0.0033 for Klf4 ChIP-qPCR, **P = 0.0047 for Etv6 ChIP-qPCR, two-tailed Student's *t*-test. **h** WISH showing that *vegfa* expression is downregulated in the somites of Klf4 binding motifs KO embryos, whereas expression in the hypochord (arrowheads) is not affected (scale bars: 0.5 mm)

corneal transparency through the repression of angiogenic factors such as VegfA[53]. In conclusion, the mechanistic relationship between Etv6 and Klf4 in regulating *vegfa* expression is compatible with a coherent feed-forward loop, a regulatory circuit where two pathways downstream of a TF control expression of the same gene: Etv6 activates *klf4*, Klf4 binds and activates *vegfa* and Etv6 binds *vegfa*[54,55] (Fig. 7). How Etv6 recruitment to *vegfa* promoter is functionally integrated to Klf4 activity in the control of *vegfa*

expression remains to be fully explored as they cannot interact directly in our study.

In addition to activating an activator of *vegfa*, Etv6 achieves *vegfa* activation by repressing a transcriptional repressor of *vegfa*, Foxo3. In humans, FOXO3 harbours tumour suppressor activity as it represses processes such as VEGFA-driven tumour angiogenesis[56]. As an example, in breast cancer, FOXO3 activation correlates with VEGFA downregulation[25]. Mechanistically, in

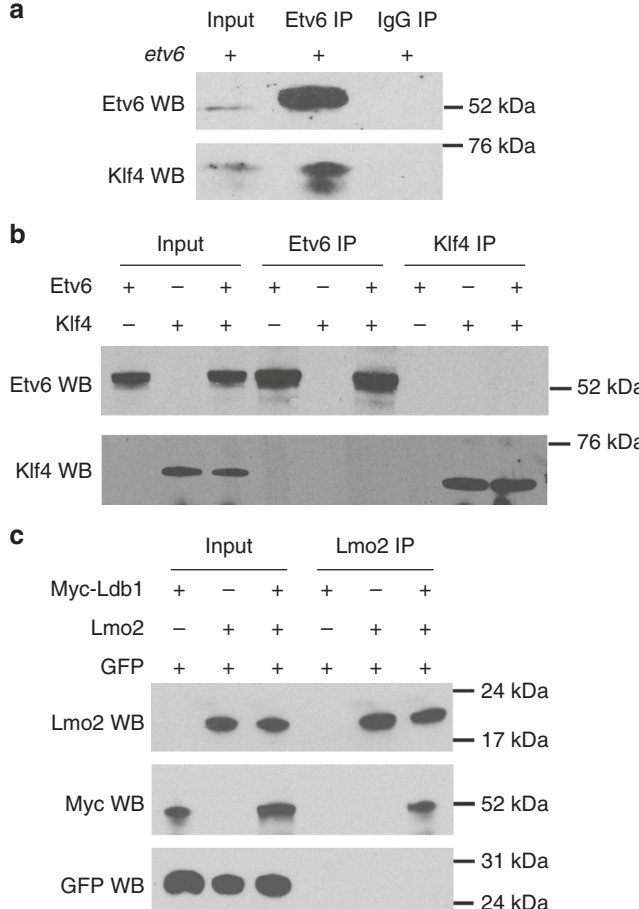

**Fig. 6** Etv6 and Klf4 interact indirectly with each other. **a** In vivo immunoprecipitation assay showing the interaction between Etv6 and Klf4. Protein extracts from the somites of stage 22 embryos overexpressing *etv6* mRNA were used for immunoprecipitation. Input lane: somite extracts from embryos injected with exogenous *etv6* mRNA; IgG immunoprecipitation: negative control. **b** In vitro immunoprecipitation assay showing that Etv6 and Klf4 do not directly interact. The Etv6 and Klf4 proteins were synthetized from mRNA using the in vitro rabbit reticulocyte lysate translation system, extracted and used in immunoprecipitation assays with Etv6 antibodies (Etv6 IP) or Klf4 antibodies (Klf4 IP). **c** In vitro immunoprecipitation assay between two proteins known to physically interact with each other, Lmo2 and Ldb1, was performed as in (**b**) with Lmo2 antibodies (Lmo2 IP). The GFP protein serves as a negative control

human cells, FOXO3 represses *VegfA* expression by binding to a Forkhead response element in the *VegfA* promoter[25]. We have confirmed that Foxo3 can bind to the conserved Forkhead binding motif (5'-GTAAACA-3')[42] in the *Xenopus vegfa* promoter region, and negatively regulates its expression in the somites as (i) overexpression of this TF results in robust *vegfa* promoter occupancy and repression of *vegfa* expression, and (ii) downregulation of *foxo3* in Etv6-deficient embryos rescues *vegfa* expression. We propose that this double negative gate[57] (Etv6 repressing Foxo3 that represses *vegfa*), together with the feed-forward loop described above, unlocks the endothelial cell fate in the lateral plate mesoderm through tight control of *vegfa* expression (Fig. 7).

Gene repression by ETV6 and Yan is mediated by the pointed (SAM) and linker domains. These domains can repress transcription independently through different mechanisms. The linker domain represses transcription by complexing with co-repressors which recruit histone deacetylases (HDACs)[14,24,58]. The pointed domain represses transcription by a mechanism that does not involve co-repressors recruiting HDACs[14]. Additionally, the pointed domain has the capacity to form homo- and hetero-typic oligomers and it has been proposed that polymerization of ETV6 could facilitate the spreading of transcriptional repression complexes along long stretches of DNA[16]. However, the formation of ETV6 polymers has not yet been demonstrated in vivo. We found a total of 9128 Etv6 ChIP-seq peaks in our analysis but only 102 of them were > 2 kb in size (Supplementary Data 1) and none were associated with DEGs. Therefore, although long stretches of DNA occupancy are present in the genome, Etv6 polymerization does not mediate gene repression in *Xenopus* somites.

In conclusion, our investigation of the functions of Foxo3 and Klf4 in *vegfa* expression in the somites unveils the foundations of a complex Etv6 GRN. It indicates that Etv6 has the capacity to regulate *vegfa* expression through the deployment of pathways involving both positive and negative regulators of gene expression and begins to unveil the complexity of *vegfa* regulation during hematovascular development. This work also further strengthens the parallel between oncogenic and developmental processes[13]. We propose that this Etv6-*vegfa* GRN is conserved through vertebrate evolution, from *Xenopus* to human, and that Etv6 uses, depending on tissue and context, particular components and/or branches of this network to ensure the correct levels of *vegfa* expression. Further investigation will establish whether Klf4, or additional members of the Klf/Sp family of TFs, are required for the regulation of some of the other Etv6 targets identified in our study and known to regulate Vegfa in the tumorigenic processes. If so, studying these molecular mechanisms could lead to new strategies for the treatment of pathological processes driven by the Etv6-*vegfa* GRN.

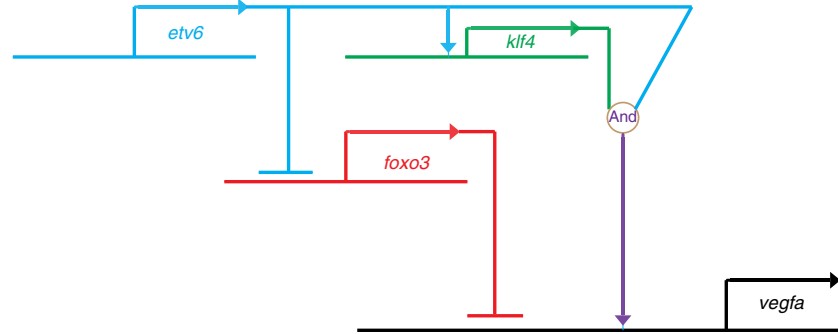

**Fig. 7** GRN summarizing the mechanisms by which Etv6 regulates the expression of *vegfa*. Etv6 positively regulates *vegfa* expression in the somites through multiple mechanisms. It represses the transcription of *foxo3*, a repressor of *vegfa* expression. In parallel, it activates the expression of *klf4*, an activator of *vegfa*. Finally, Klf4 is required for Etv6 recruitment to the *vegfa* promoter to activate its expression. In conclusion, a network of positive and negative inputs from Etv6 is required for transcriptional activation of *vegfa* in the somites

## Methods

**Embryo manipulation**. Animal experiments were performed according to UK Home Office regulations under the appropriate project licence and approval of the University of Oxford Animal Welfare and Ethical Review Body. Wild type (WT) *Xenopus laevis* embryos were injected in 0.35 × MMR (1×Marc's Modified Ringer: 100 mM NaCl, 2 mM KCl, 1 mM MgCl₂, 2 mM CaCl₂, 5 mM HEPES, pH 7.5. Prepare a 10 x stock, and adjust pH to 7.5) supplemented with 3% Ficoll. After 6 h, embryos were washed and cultured in 0.1×MMR at 17–19 °C[59] until they were collected for analysis. Embryos were staged according to Nieuwkoop and Faber[60]. The somites of stage 22 embryos were manually dissected in 0.35 × MMR using forceps, these explants also contained the hypochord, neural tube, notochord, and some dorsal endoderm.

**Whole-mount in situ hybridization**. Whole-mount in situ hybridization (WISH) was performed using a standard protocol[59]. Briefly, embryos were fixed overnight in MEMFA (0.1 M MOPS, pH 7.4; 2 mM EGTA; 1 mM MgSO₄; 3.7% for-maldehyde) and then dehydrated to 100% methanol and stored at −20 °C. Embryos were rehydrated through a methanol gradient in PBST (75%, 50%, 25%, and 0%) and then incubated in a 5% formamide, 0.5% SSC, and 10% H₂O₂ solution to remove embryo pigmentation. Bleaching solution was washed with PBST (3 × 5 min). At this point, stage 39 embryos were treated with 8 µg/ml Proteinase K for 1 min and then washed in glycine (2 mg/ml in PBST, 3 × 5 min) to inactive Pro-teinase K (younger embryos do not require Proteinase K treatment). Embryos were then washed in PBST (3 × 5 min) and incubated in 0.1 M Triethanolamine-HCl (TEA, pH 7.9, 5 min) before treating with acetic anhydride (0.25% v/v in TEA, 2 × 5 min). Embryos were rinsed with PBST (3 × 5 min) and then pre-hybridized in hybridization buffer (50% formamide, 5×SSC, 1 mg/ml Torula RNA, 100 µg/ml Heparin, 1× Denhardt's, 0.1% Tween and 5 mM EDTA) for 6 h at 67 °C before incubating in 0.5 ml hybridization buffer containing DIG-labelled probes (1 µg/ml each) for 16 h at 67 °C. After hybridization, embryos were sequentially washed in: (i) 50% formamide/5xSSC, 67 °C, 10 min; (ii) 25% formamide/2xSSC, 65 °C, 10 min; (iii) 12.5% formamide/2xSSC, 63 °C, 10 min; (iv) 2×SSC, 0.1% Tween, 61.5 °C, 10 min; (v) 0.2×SSC, 0.1% Tween, 60 °C, 30 min. After washing in PBST (3 × 5 min) and MAB buffer (1 M Maleic Acid, 0.15 M NaCl, pH 7.5; 3 × 5 min), embryos were blocked for 6 h in MAB blocking solution (MAB buffer + 2% Boehringer Blocking Reagent). After blocking, embryos were incubated overnight at 4 °C in MAB blocking solution containing anti-DIG-AP antibody (1/2,000 dilution). Next, embryos were washed in MAB buffer (6 × 1 h) while rocking gently, followed by 3 × 5 min washes in AP buffer (0.1 M Tris, pH 9.0; 50 mM MgCl₂; 0.1 M NaCl; 0.1% Tween) and then stained in BM purple solution (diluted 1:1 with AP buffer) in the dark. After staining, embryos were fixed in MEMFA for 1 h, dehydrated in 100% methanol and stored at -20 °C. Before photography, embryos were cleared in benzylbenzoate:benzyl alcohol (2:1). All procedures were performed at room temperature unless otherwise indicated.For probe details see Supplementary Table 3.

**MO and mRNA for injection**. MOs (morpholino antisense oligonucleotide) were obtained from GeneTools LLC (Corvallis, OR). *Etv6* MO targeting both *etv6.L* and *etv6.S* was previously published[12]. A *klf4* MO targeting both *klf4.L* and *klf4.S* was designed (Fig. 4e). As single MO blocking both *foxo3.L* and *foxo3.S* could not be designed (Supplementary Figure 5a), two MOs, one targeting *foxo3.L* (*foxo3.L* MO) and the other targeting *foxo3.S* (*foxo3.S* MO), were co-injected in a 1:1 ratio in order to generate Foxo3-deficient embryos (Supplementary Figure 5c). Every MO was titrated in order to determine the optimal concentration for embryo injection. MO sequences are as indicated in Supplementary Table 4.

A two-step Golden Gate assembly method using the Golden Gate TALEN and TAL effector kit 2.0 (Addgene) was used to construct the TALEN plasmids containing the homodimer-type FokI nuclease domain[61]. TALEN sequences were designed using the online design tool, Mojo Hand (http://www.talendesign.org/). TALEN mRNA for injection was generated by linearizing the plasmids with NotI and transcribing with SP6 RNA polymerase using the mMESSAGE mMACHINE Kit (Ambion). TALEN mRNAs were injected at the 1-cell stage of development. To determine the mutagenesis caused by the TALENs, DNA was extracted from the somites of stage 22 TALEN-injected embryos and the genomic DNA fragment containing the TALEN binding sites was amplified and Sanger sequenced (Fig. 4f, Supplementary Figure 11).

To make mRNA for injection, the full-length coding sequence for *X. laevis etv6* and *foxo3*, including the sequences targeted by the MOs (Supplementary Table 5), and *klf4*, were sub-cloned into *pBUT3-HA* vector (*pBUT3-etv6-HA, pBUT3-foxo3-HA* and *pBUT3-klf4-HA*). Etv6-HA, *foxo3*-HA and *klf4*-HA mRNAs were synthesized using mMESSAGE mMACHINE T3 Kit (Ambion). mRNAs were injected at the 2-cell stage of development.

**Antibodies and western blot analysis**. Polyclonal anti *X. laevis* Etv6 antibodies were generated in collaboration with NovoPro Bioscience Inc. (Shanghai, China). In brief, the amino acid sequence of *X. laevis* Etv6 (GeneBank number EU760352) was analyzed using a proprietary algorithm, NovoFocus™ antigen design, to identify epitopes with good hydrophilicity, surface probability and high antigenic index. Based on this information, three epitopes with little or no conservation with

other proteins, including ETS TFs, were selected (Supplementary Figure 1a). Short peptides corresponding to these epitopes were synthesized, conjugated to keyhole limpet hemocyanin (KLH) and used in immunizations at a purity ≥90%. Two rabbits were immunized with each peptide and six polyclonal antibodies (Etv6-1a, -1b, -2a, -2b, -3a, -3b) were affinity-purified after 5–6 rounds of immunization. Each antibody had an ELISA titre ≥1:50,000 against the peptide antigen. The specificity of the antibodies was verified by Western Blot analyses.

*X. laevis* Klf4 protein was detected using an anti-Klf4 antibody (Abcam, ab106629) at a 1:1000 dilution. HA-tagged protein was detected with anti-HA antibody (Santa Cruz Biotechnology, HA-probe (Y-11)-G) at a 1:500 dilution. Histone H3 protein was detected using anti-Histone H3 (Abcam, ab1791) antibody at a 1:100,000 dilution. GFP protein was detected using anti-GFP antibody (Millipore, MAB3580) at a 1:5000 dilution. Lmo2 antibody (Abcam, ab72841) was used in the immunoprecipitation assays. Myc-tagged protein was detected with anti-Myc antibody (Santa Cruz Biotechnology, sc-40) at a 1:1000 dilution. Foxo3 antibody (Thermo Fisher Scientific, Catalog number: 720128) was used for ChIP-qPCR. Clean-blot IP detection kit (Thermo Fisher Scientific, Catalog number: 21232) was used as a secondary antibody in the Western blot for the immunoprecipitation experiments and was used at a 1:200 dilution. Protein extracts were prepared by lysing 10 somites from stage 22 embryos in 50 µl of oocyte lysis buffer (20 mM Tris-HCl pH 7.5, 10 mM EDTA, 1%Triton X-100, 1 mM β-glycerophophate, 3 mM Pefabloc and 3 mM DTT. Add protease inhibitors just before use), and centrifuging at 13,000 rpm at 4 °C for 10 min. An equal volume of 2xRSB (reducing sample buffer, 6% β-mercaptomethanol, 6% SDS, 0.6% bromophenol blue and 4% glycerol) was added to the supernatants and boiled at 100 °C for 5 min and stored at −20 °C. 20 µl of protein extract (2 somite explants equivalent) was ran in SDS-PAGE gels and transferred to 0.2 µm nitrocellulose. Membranes were blocked with 1% milk in PBST, and probed with appropriate primary antibodies and horseradish peroxidase-conjugated secondary antibodies. Uncropped blots for Figs. 4e and 6a–c and Supplementary Figures 1b–c, 5b, 11b have been provided in Supplementary Figures 14 and 15.

**ChIP-seq**. Etv6 ChIP-seq was performed using a standard procotol[62,63] with some modifications. Briefly, the somites of 40 stage 22 embryos were dissected, homo-genized, using a pestle, in 1 ml ice-cold PBS containing protease inhibitors and 20 mM sodium butyrate, and centrifuged at 1300 rpm for 5 min at 4 °C. Pellet was re-suspended and cross-linked in 2 mM EGS (ethylene glycol bis(succinimidyl suc-cinate)) for 20 min at RT, followed by fixation in 1% formaldehyde for 20 min at RT and gentle rocking before treating with glycine (final concentration: 125 mM) for 5 min. Samples were centrifuged (1,300 rpm, 5 min, 4 °C) and pellets rinsed twice with cold PBS, lysed and chromatin sonicated to an average size of ~200 bp using Covaris S220 (Duty factor: 10%; Cycle per burst: 200; Intensity: 5; Duration time: 300 s; Cycle: 3). Samples were centrifuged for 10 min at 14,000 × g and supernatant recovered. After saving 1% of each sample as input, Triton X-100 (final concentration: 1%) was added to the supernatant and incubated overnight at 4 °C with slow shaking in the presence of anti-Etv6 antibody already bound to Dyna-beads protein A. Beads were washed in RIPA buffer (4 °C, 10 × 5 min) followed by 2 washes (5 min) in TEN buffer (10 mM Tris-HCl, 1 mM EDTA, 150 mM NaCl; 4 °C). Complexes were eluted in 1% SDS elution buffer (50 mM Tris-HCl, 1 mM EDTA and 1% SDS) by shaking the Dynabeads for 15 min at 65 °C in a thermo-mixer set to 1000 rpm. Eluted samples were centrifuged at full speed for 1 min at RT. The supernatants and the reserved input samples were de-crosslinked by adding 1/20 volume of 5 M NaCl and incubating at 65 °C overnight. After treating with RNase A (final concentration: 0.2 mg/ml, 1 h at 37 °C) and Proteinase K (final concentration: 0.2 mg/ml, 3 h at 55 °C), DNA was extracted using phenol:chloro-form:isoamyl alcohol purification followed by ethanol precipitation. Immuno-precipitated DNA and input DNA were quantified using Qubit Fluorometer and sequencing libraries were constructed with 1 ng of DNA using NEBNext Ultra DNA Library Prep Kit (NEB#E7370) for Illumina sequencing. DNA was sequenced (~15–38 million paired reads/library, 2 × 40 bp read length) using NextSeq 500. Sequencing was performed on three independent biological replicates with corre-sponding inputs as control.

Raw sequence reads were checked for base quality, trimmed and filtered to exclude adapters using *Trimmomatic* (Version 0.32)[64], and then mapped to the *X. laevis* V9.1[40] with BWA version 0.7.12. Peaks were analyzed using MACS2 peak calling software[65] at default thresholds, with the input samples as control for each replicate. The package DiffBind (Bioconductor package)[29] was employed to identify consistent peaks between replicates using the function dba.peakset with the parameter minOverlap = 3. The consensus peak output of DiffBind was generated by adding and merging the three replicate peak sets. Peaks located in TSS regions were identified using Homer software and used to perform TFs binding site analysis with Homer software (findMotifsGenome.pl)[30]. Examples of the Etv6 peaks for individual replicates are shown in Supplementary Figure 13. For TF binding site prediction in the *vegfa* promoter region (−2 kb to +1 bp) and the Etv6 peak region (−552 to −244 bp upstream of *vegfa.L* TSS), DNA sequences were analyzed using Jaspar[41]. ChIP-seq data were visualized on the Integrative Genome Viewer (IGV).

**RNA-seq**. Total RNA was extracted from somites dissected from 20 WT or Etv6-deficient embryos at stage 22. Triple biological replicates were generated. Indexed

libraries were constructed with 1 μg of total RNA using the KAPA Stranded RNA-seq Kit with RoboErase (KK8483) and NEBNext Mutiplex Oligos (NEB#E7500S) for Illumina sequencing. DNA was sequenced (∼60–105 million paired reads/library, 2 × 75 bp read length) using NextSeq 500.

Sequenced reads were checked for base qualities, trimmed and filtered to exclude adapters using *Trimmomatic* (Version 0.32)[64] and then mapped to the *X. laevis* V9.1[40] using *STAR*[66] with default parameters. Aligned read features were counted using Subread tool: featureCounts method (version 1.4.5-p1). Differential gene expression analysis was carried out using EdgR (Bioconductor Package)[67]. For this analysis, samples were CPM (counts per million)-normalized in order to filter out genes with low counts and, then, TPM (transcript per kilobase million)-normalized for differential expression analysis. CPM normalization accounts for library size differences between samples and produces normalized values that can be compared on an absolute scale.

An ANOVA-like test analysis[68] was performed between WT and MO samples, using the generalized linear model[69] followed by Estimates Dispersion, fitted with negative binomial model and estimates Generalized linear model likelihood ratio. Genes with a false discovery rate (FDR) above 0.05 were filtered out. The functional annotation (gene ontology analysis) of Etv6-regulated transcriptome was performed using DAVID Bioinformatics Resources[70].

**qPCR**. qPCR (quantitative PCR) was performed using Fast SYBR Green Master Mix (Thermo Fisher Scientific, Catalog number:4385612) and StepOne Real-Time PCR system. For RT-qPCR, cDNA was made from 1 μg of total RNA and relative expression levels of each gene were calculated and then normalized to *odc1* gene. Details of primer sequences for RT-qPCR and ChIP-qPCR are indicated in Supplementary Tables 6 and 7, respectively.

**Luciferase assay**. The *vegfa* promoter region containing a conserved Foxo3 binding site as well as the *klf4* and *foxo3* promoter regions containing Etv6 peaks were amplified from *X. laevis* genomic DNA by PCR using primers described in Supplementary Table 5. Two-step PCR-mediated mutagenesis was used to delete the Foxo3 binding site in the *vegfa* promoter fragment. All the produced sequences were subcloned into the pGL4.10 vector (Promega). 100 pg reporter plasmid DNA together with 5 pg Renilla luciferase construct pRL1-TK (Promega) and *etv6* MO or *foxo3* mRNA, were co-injected into 2-cell stage embryos. Somites were dissected at stage 22, and assayed for luciferase activity using the Dual Luciferase Reporter Assay Kit (Promega).

**In vitro protein synthesis**. *Xenopus* Etv6-HA, Klf4-HA, Ldb1-Myc, mouse Lmo2, and GFP proteins were synthesized from mRNAs (Supplementary Table 8) using the in vitro rabbit reticulocyte lysate translation system (Promega) according to the manufacturer's instructions.

**Co-immunoprecipitation assay**. Embryos were injected with *etv6-HA* mRNA at the 2-cell stage and grown to stage 22, when proteins were extracted by lysing 30 somites in 300 μl of lysis buffer (10 mM Tris-HCl pH 7.5, 100 mM NaCl, 0.5% Triton X-100, and 2.5 mM MgCl₂, protease inhibitors added). After sonication for 20 min at 4 °C using Bioruptor Plus (30 s on and 30 s off for each round of sonication), the extracts were centrifuged at 13,000 rpm at 4 °C for 10 min. After saving 1% of each sample as input, the supernatants were incubated overnight at 4 °C with slow shaking in the presence of anti-Etv6 or IgG antibody already bound to Dynabeads protein A. After washing three times in lysis buffer containing protease inhibitors, protein complexes were eluted in 1% SDS elution buffer by shaking the Dynabeads for 15 min at 65 °C in a thermomixer set to 1000 rpm. Eluted proteins were separated by SDS-PAGE and transferred onto a 0.2 μm nitrocellulose membrane for western blotting. For in vitro co-IP, synthesized proteins were first incubated in lysis buffer for 1 h at 4 °C. After saving 10% of each sample as input, the lysates were used for immunoprecipitation as described above.

**Statistical analysis**. In RT-qPCR and ChIP-qPCR experiments, error bars represent Standard Error of Mean (±SEM). The data shown summarize the results of three biological replicates. Two-tailed Student's *t*-test was performed (*$p < 0.05$; **$p < 0.01$; ***$p < 0.001$).

WISH images and numbers shown in Figures are from one experiment and are representative of three biological replicates.

No statistical methods were used to predetermine the sample size.

No data or animal have been excluded from this study.

*Xenopus laevis* embryos were allocated to experimental groups on the basis of different treatments and randomized within the given group. Investigators were not blinded to the group allocation during experiments and outcome assessment.

**Reporting summary**. Further information on experimental design is available in the Nature Research Reporting Summary linked to this article.

## Data availability

All ChIP-seq and RNA-seq datasets have been deposited in the Gene Expression Omnibus (GEO) database under accession code GSE115225.

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

## Acknowledgements

We are grateful to Dr. Arif Kirmizitas for providing plasmids for the synthesis of tagged proteins and for advice on co-IP experiments. This work was supported by the UK Medical Research Council (MRC, MC_UU_12009/9) and Biotechnology and Biological Sciences Research Council (BBSRC, BB/M001938/1).

## Author contributions

L.L., R.P., A.C. and C.P. designed the experiments. L.L. performed all experiments, prepared the figures and wrote the manuscript. A.C. designed and generated TALEN and designed peptides for Etv6 antibodies production. R.R. performed bioinformatics analyses. R.P., A.C. & C.P. analyzed the data and revised the manuscript. All authors read and approved the final manuscript.

## Additional information

**Competing interests:** The authors declare no competing interests.

