## [Peer Review File · Nature Communications]

Reviewers' Comments:

Reviewer #1:

Remarks to the Author:

This manuscript explores the transcription factors (TFs) and mechanisms regulating expression of Vegfa in cells critical for hematopoiesis and angiogenesis, specifically in stage 22 somites of *Xenopus*. The authors build on their previous work implicating the TF ETV6 in positive regulation of Vegfa expression. They map the binding sites of ETV6 genome-wide, identify genes differentially expressed after ETV6 knockdown, and use those two datasets to identify likely regulators among the ETV6 target genes. Based on their own results and information in the prior literature, they focused on two TFs, FOXO3 implicated in repression of Vegfa and KLF4 implicated in positive regulation. Through carefully executed biochemical and genetic experiments, the authors show that ETV6 regulates Vegfa via FOXO3 using a double negative gate (ETV6 represses Foxo3 whose product represses the Vegfa gene). Another series of thorough experiments reveal a coherent feed-forward loop for the positive regulation of Vegfa by ETV6 via KLF4. These experiments reveal a complex gene regulatory network for Vegfa and they provide novel insights both into how ETV6 regulates target genes (positively and negatively).

The experiments were conducted carefully and follow best practices. The biological insights were examined thoroughly in multiple ways, e.g. with knockouts of genes, repression of translation via morpholinos, and genomic editing of binding sites for the TFs. The presentation is clear and concise, and the Discussion places the results in a larger context, including relevance to certain cancers.

I have only a few minor comments for the authors to consider.

(1) The authors describe how the antibodies against ETV6 were developed, and they show that this protein is a target for the antibody both in Western blots and in immunoprecipitation. They only show a small portion of the Western blots. It would be good to also address the selectivity of the antibody - is the ETV6 band the only band, or predominant band in the assay?

(2) The authors conducted ChIP-seq experiments on three replicates, and used the replicates to find consistent peaks. I wondered how they had found consistent peaks - was it by a simple overlap, or did they use IDR (irreproducible discovery rate)? In the Methods they tell us that they used DiffBind to identify the consistent peaks. That information would be good to include in the Results, and they could provide more information in the Methods about how exactly DiffBind was used on the triplicate determinations to establish consistency.

(3) The example ChIP-seq data in the figures show a single track for ETV6 binding. Are the signals plotted for the mapped reads pooled across the replicates? Showing the signals for individual replicates might provide a convincing case for reproducibility.

(4) Fig. 2c legend: The authors call this a smear plot, and that is the name of some tools for making the graph. However, it looks like an MA plot, which is a more conventional name (magnitude of the difference versus amplitude of the signal).

(5) The authors use the unit "CPM" for their transcript measurements. They should explain what this is, and how it compares with TPM and FPKM.

I support the transparency of Nature Communications' review system, and thus I sign my review.

Ross Hardison

Reviewer #2:

Remarks to the Author:

This manuscript by Li and colleagues explores the molecular mechanisms by which the transcription factor ETV6 regulates *vegfa* expression during embryonic development using *Xenopus* as a model system. Through a series of elegant experiments, the authors demonstrate that ETV6 controls the transcription of *Foxo3*, a repressor of *vegfa*, and the transcription of *klf4*, an activator of *vegfa*. They further demonstrate that through KLF4, ETV6 is recruited to the promoter of *vegfa*, leading to an additional layer of regulation in this gene regulatory network. Overall, this is a very well written manuscript, easy to follow with convincing data. The results presented further unravel the transcriptional regulation of *vegfa*, a complex and critical process during blood cell emergence and vascular development.

To complement the data presented, it would be interesting to show whether ETV6 interacts directly with KLF4 (via *in vitro* co IP approaches) or whether scaffolding or other proteins are likely necessary for the recruitment of ETV6 to the promoter region of *vegfa*.

Similarly, it would be nice to show (via luciferase-type transcription assays with mutated and wt promoter region) that indeed FOXO3 binds to the newly identified putative site in the *vegfa* promoter and negatively regulates transcription.

Is the control of *vegfa* expression by this ETV6 GRN unique to the somites at stage 22 or is this also seen elsewhere during development?

Minor comments:

Line 170: "Surprisingly, the majority of *Etv6* target genes (303/482, 63%) were downregulated in the somites of *etv6*-deficient embryos". Given that "only" 63% of the genes are downregulated, it might be more appropriate to write "a large fraction" or "around two third" rather than "the majority".

Line 171-172: "strongly indicating that *Etv6*, typically considered as a transcriptional repressor, acts mainly as an activator of gene". Rather than "mainly", it might be more appropriate to use the word "also".

Reviewer #3:

Remarks to the Author:

Here, Lei Li and coworkers examine the role of the transcription factor *Etv6* in regulation of *Vegfa* expression during *Xenopus laevis* development. Main findings are that *Etv6* suppresses expression of the negative regulatory *Foxo3* while it enhances expression of the positive regulatory *Klf4*. Moreover, *Klf4* recruits *Etv6* to the *Vegfa* promoter. The fact that TFs act alone or in complex with other factors to both enhance and repress gene regulation is of course well established already also for *Etv6* and was suggested by Cia-Uitz et al., to explain the mode of action of *Etv6* in regulation of *Vegfa* expression in their 2010 *Dev Cell* paper. Moreover, reports on *Foxo3* as a repressor and *Klf4* as an inducer of *Vega* transcription have been published. The role of *Etv6* in repression and activation of *Foxo3* and *Klf4*, respectively, presented here, is new. Also, not as much is known about gene regulation in *X. laevis*, which is the preferred model of the Porcher laboratory, as in other genetic models. The study is well presented. Overall, however, comprehensive aspects of *Vegfa* gene regulation and *Etv6*/TF mode of action are missing.

1. To map *Etv6* binding sites in the *X. laevis* genome, anti-*Etv6* antibodies were raised for DNA immunoprecipitation followed by sequencing. The authors list a few genes showing the highest enrichment of *Etv6* in their transcription start site (*rimklb* and *Xelaev18005411m*). *Vegfa* is not in

this list and the two genes are just mentioned which is not very meaningful. The authors go on to conclude that Etv6 peaks in the TSS analysis do not contain ETS family binding motifs but show motifs compatible with binding of other TFs and that Etv6 exerts its gene regulatory effect by complex formation with a range TFs. Here, the authors describe only the TSS regions which make up about 25% of the Etv6 peaks in the ChipSeq. Why not analyze the remaining Etv6 peaks as well? Are findings only relevant to the TSS region?

2. To obtain a more comprehensive analysis of Vegfa gene regulation, it would have been interesting to see a map all known TF motifs in the Vegfa gene. Some of these TFs would be Etv6 regulated, as mentioned in the first part of the discussion, others would not. The limitation of this approach would be that distant enhancers would not be detected. However, now, the analysis is focused rather narrowly to Foxo3 and Klf4 based on that they bind to the Vegfa TSS and that their expression is regulated by Etv6. Would TF genes need to be repressed/activated by Etv6 for it to engage in TF complexes regulating Vegfa expression?

3. The authors emphasize that Klf4 recruits Etv6 to the Vegfa promoter as enrichment of Etv6 detected by ChipSeq was decreased upon deletion of Klf4 binding motifs in the Vegfa promoter. Would not Foxo3 also recruit Etv6 to the Vegfa promoter?

Minor

4. The comment in the end of the abstract on the potential usefulness of the presented data in new strategies for treatment of cancer, should be removed. Clearly, this study does not have any bearing on tumor therapy and to add such statements as is often done, just takes credit away.

5. In the intro, second para, it says that knockout of VegfA receptor (Vegfr1,2 or 3) genes results in early lethality – VEGFR3 is not a receptor for VEGFA, so please adjust.

6. In the results part under “Establishing the transcriptome regulated by Etv6 in the somites”, please motivate why stage 22 embryos were used for the general readership.

Reviewer #4:

Remarks to the Author:

The authors performed Etv6 ChIP-seq using somites of the stage 22 frog embryos to understand the mechanisms by which Etv6 regulates Vegfa expression. While Etv6 is believed to repress its target genes, they found Etv6 to both activate and repress its target genes. While repressed genes have an ETS motif in their promoter regions, activated genes don't have an ETS site. Importantly, they identified potential activator as well as repressor of Vegfa expression. For example, Klf4, a presumed activator of Vegfa is activated by Etv6, and the authors propose a model that Etv6 activates Klf4, which in turn recruits Etv6 to the Vegfa promoter region, and activates Vegfa expression. While this study is the first to perform Etv6 ChIP-seq and suggest Etv6 can function both as an activator and repressor of the target genes, data sometimes is indirect and incomplete in supporting their argument. Particularly, the study falls short of providing mechanistic insights of how Etv6 can function both as an activator and repressor at the same time. Vegfa expression in somites does not necessarily reflect direct linear causal relationship of the various manipulations.

Overall, target gene validation is not optimal. Peak enrichment on the target site is not sufficient to establish that the identified genes are indeed direct target genes. All potential target genes should be validated by taking the potential binding regions and performing luciferase reporter assays, at least.

The mechanisms by which Etv6 functions as an activator or repressor is not clear. For example, the idea they like the readers to believe is that Etv6 represses its target genes through an ETS site. I would like to see that the authors take several repressed genes and perform luciferase assay to show that indeed these TSS regions of the repressed genes are responsible for Etv6 repression. Also, I would like to see the mutational analysis of the ETS site(s) of the target genes, and when mutated, the reporter expression in turn goes up. As for the activated genes, I also want to see the similar types of data.

The authors propose that Klf4 recruits Etv6 to activate Vegfa expression. I would like to see the direct interaction between Klf4 and Etv6.

Can the authors show that Klf4 can rescue Etv6 MO effect on Vegfa, Tal1, and Runx1 expression?

I cannot seem to find the data showing that Etv6 expression is not affected by either Foxo3a or Klf4 MO treatment, which is critical.

Responses to Reviewers

We thank the four Reviewers for their positive and constructive comments and helpful suggestions.

Reviewer #1 (Remarks to the Author):

This manuscript explores the transcription factors (TFs) and mechanisms regulating expression of Vegfa in cells critical for hematopoiesis and angiogenesis, specifically in stage 22 somites of *Xenopus*. The authors build on their previous work implicating the TF ETV6 in positive regulation of Vegfa expression. They map the binding sites of ETV6 genome-wide, identify genes differentially expressed after ETV6 knockdown, and use those two datasets to identify likely regulators among the ETV6 target genes. Based on their own results and information in the prior literature, they focused on two TFs, FOXO3 implicated in repression of Vegfa and KLF4 implicated in positive regulation. Through carefully executed biochemical and genetic experiments, the authors show that ETV6 regulates Vegfa via FOXO3 using a double negative gate (ETV6 represses Foxo3 whose product represses the Vegfa gene). Another series of thorough experiments reveal a coherent feed-forward loop for the positive regulation of Vegfa by ETV6 via KLF4. These experiments reveal a complex gene regulatory network for Vegfa and they provide novel insights both into how ETV6 regulates target genes (positively and negatively).

The experiments were conducted carefully and follow best practices. The biological insights were examined thoroughly in multiple ways, e.g. with knockouts of genes, repression of translation via morpholinos, and genomic editing of binding sites for the TFs. The presentation is clear and concise, and the Discussion places the results in a larger context, including relevance to certain cancers.

I have only a few minor comments for the authors to consider.

(1) The authors describe how the antibodies against ETV6 were developed, and they show that this protein is a target for the antibody both in Western blots and in immunoprecipitation. They only show a small portion of the Western blots. It would be good to also address the selectivity of the antibody - is the ETV6 band the only band, or predominant band in the assay?

Uncropped Western Blots were presented in Supplementary Figure 9, now Supplementary Figure 14.

In brief, when testing their specificity by Western Blot analysis on whole protein extracts, antibodies Etv6-1b, -2a and -2b cross-reacted, in addition to Etv6, with proteins of higher and lower molecular weight (Supplementary Figure 14, see panels corresponding to Supplementary Figure 1b). However, when tested for their capacity to immunoprecipitate Etv6, Etv6-2a was more efficient than the other antibodies and was highly selective as only one band, corresponding to Etv6, was detected by Western Blot (Supplementary Figure 14, see panel corresponding to Supplementary Figure 1c). This is the reason why we selected Etv6-2a for the ChIP-seq experiments.

(2) The authors conducted ChIP-seq experiments on three replicates, and used the replicates to find consistent peaks. I wondered how they had found consistent peaks - was it by a simple overlap, or did they use IDR (irreproducible discovery rate)? In the Methods they tell us that they used DiffBind to identify the consistent peaks. That information would be good to include in the Results, and they could provide more information in the Methods about how exactly DiffBind was used on the triplicate determinations to establish consistency.

Consistent peaks were determined using the consensus Overlap function in the package DiffBind. More specifically, the function `dba.peakset` with the parameter `minOverlap=3` was employed to only identify the

peaks present in all 3 ChIP-seq replicates. Thus, the list of consensus peaks only includes peaks present in the three replicates. As suggested by the Reviewer, this information is now included in the Methods (page 27).

The calculations used in DiffBind do not include low-confident binding sites and noise, as long as a sufficient number of replicates (at least 3) are processed to properly power the analysis. Therefore, as the analysis was performed on 3 biological replicates, IDR was not required. The statistical analysis was conducted with DiffBind and the underlying package DESeq2.

(3) The example ChIP-seq data in the figures show a single track for ETV6 binding. Are the signals plotted for the mapped reads pooled across the replicates? Showing the signals for individual replicates might provide a convincing case for reproducibility.

The consensus peaks shown in the examples (Figures 1c, 3a, 4a, 5a) were generated using DiffBind. The single peak set output results from merging the peak sets generated from the three replicates, not from pooling them. Peaks are merged if they overlap by ≥ 1 bp. Thus, the ETV6 ChIP signals plotted for each peak represent the peaks across the three ChIP-seq replicates. The option of keeping all peaks in a single track, i.e. not merged, raises questions such as how to count reads that overlap multiple peaks. The reproducibility of the ChIP-seq peaks can be attested by the tracks of each individual replicate, as shown in Supplementary Figure 13.

(4) Fig. 2c legend: The authors call this a smear plot, and that is the name of some tools for making the graph. However, it looks like an MA plot, which is a more conventional name (magnitude of the difference versus amplitude of the signal).

Figure 2c is indeed an MA plot. We apologize for the mistake. The figure legend has been corrected.

(5) The authors use the unit "CPM" for their transcript measurements. They should explain what this is, and how it compares with TPM and FPKM.

CPM (counts per million) is basically depth-normalized counts. CPM mapped reads are counts scaled by the total number of fragments sequenced times one million. This unit is related to the FPKM without length normalization and a factor of 10^3 .

TPM (transcripts per kilobase million) indicates counts per length of transcript (kb) per million reads mapped. This unit is length-normalized counts (and then normalized by the length-normalized values of the other genes). Thus, TPM is a measurement of the proportion of transcripts in a pool of RNA and accounts for both sequencing depth and gene length.

RPKM/FPKM (reads/fragments per kilobase of exon per million reads/fragments mapped) is similar to TPM as it accounts for both sequencing depth and gene length, however, is not recommended for differential expression analysis because the count values output is not comparable between samples. Using RPKM/FPKM normalization, the total number of counts for each sample will be different and, therefore, cannot compare the normalized counts for each gene equally between samples. In contrast, TPM-normalized counts have the same TPM-normalized counts per sample and, therefore, the normalized count values are comparable both between and within samples.

CPM normalization accounts for library size differences between samples and produces normalized values that can be compared on an absolute scale (e.g. for filtering). As tools for differential expression analysis are

comparing the counts between sample groups for the same gene, length does not need to be accounted for. Thus, CPM-normalized samples can be used for differential expression analysis.

We used the EdgeR package for differential expression analysis that recommends CPM (or TPM) for filtering out genes with low counts and, then, TPM normalization for the actual differential expression analysis. This is now indicated in the Methods (page 28).

I support the transparency of Nature Communications' review system, and thus I sign my review.

Ross Hardison

--

Reviewer #2 (Remarks to the Author):

This manuscript by Li and colleagues explores the molecular mechanisms by which the transcription factor ETV6 regulates *vegfa* expression during embryonic development using *Xenopus* as a model system. Through a series of elegant experiments, the authors demonstrate that ETV6 controls the transcription of *Foxo3*, a repressor of *vegfa*, and the transcription of *klf4*, an activator of *vegfa*. They further demonstrate that through KLF4, ETV6 is recruited to the promoter of *vegfa*, leading to an additional layer of regulation in this gene regulatory network. Overall, this is a very well written manuscript, easy to follow with convincing data. The results presented further unravel the transcriptional regulation of *vegfa*, a complex and critical process during blood cell emergence and vascular development.

To complement the data presented, it would be interesting to show whether ETV6 interacts directly with KLF4 (via *in vitro* co IP approaches) or whether scaffolding or other proteins are likely necessary for the recruitment of ETV6 to the promoter region of *vegfa*.

As requested by the Reviewer (and also by Reviewer 4), we have performed co-IP experiments to test whether Klf4 directly interacts with Etv6 or requires auxiliary proteins to recruit Etv6 to the *vegfa* promoter.

We first performed an *in vivo* co-IP experiment on stage 22 somitic protein extracts. As immunoprecipitation on endogenous Etv6 protein did not reveal interaction with Klf4 (likely due to the low level of expression of this TF), we overexpressed exogenous Etv6 protein in developing embryos and then assayed its capacity to co-immunoprecipitate endogenous Klf4 protein. This experiment indicated that Klf4 co-purifies with Etv6 (Figure 6a, and text page 16/17).

To test if this interaction occurs directly, Etv6 and Klf4 proteins were then produced by *in vitro* translation using the rabbit reticulocyte lysate system and then used in co-IP assays. In these conditions, these two proteins did not co-immunoprecipitate (Figure 6b, and text page 17). In contrast, as a positive control, two proteins known to directly interact, namely Lmo2 and Ldb1 (Lahlil *et al.*, *MCB*, 2004), did co-IP in the assay (Figure 6c). This result indicates that, *in vitro*, there is no direct physical interaction between Etv6 and Klf4.

Taken together, these experiments show that Klf4 and Etv6 are part of multiprotein complexes and suggest that Klf4 recruits Etv6 to the *vegfa* promoter through auxiliary proteins.

The co-IP protocols have been added to the Methods section (page 30/31).

Similarly, it would be nice to show (via luciferase-type transcription assays with mutated and wt promoter region) that indeed FOXO3 binds to the newly identified putative site in the *vegfa* promoter and negatively regulates transcription.

We thank the Reviewer for this suggestion. We have performed both Foxo3 ChIP experiments and luciferase assays to demonstrate the functional binding of Foxo3 to the *vegfa* promoter.

First, binding of Foxo3 to its conserved binding site in the *vegfa* promoter has been confirmed by ChIP-PCR on stage 22 somitic material and is shown in Figure 3g (see also text page 13). In line with the very low level of *foxo3* expression in the somites of WT embryos (in fact, hardly detectable by WISH, Figure 3c), levels of Foxo3 occupancy are significantly increased in *Etv6*-deficient embryos (where *foxo3* expression is upregulated, see Figure 3c, d) and upon expression of exogenous *foxo3* mRNA (Figure 3g).

Additionally, as suggested by the Reviewer, we have tested the transcriptional activity of the *vegfa* promoter sequences containing Foxo3 binding site through luciferase assays. These were performed *in vivo*, upon injection of luciferase constructs in 2-cell stage embryos and luciferase assays conducted on stage 22 somitic material. The results are presented in new Supplementary Figure 7 (and text page 13). The ability of the sequence containing the Foxo3 binding site to activate transcription was repressed when Foxo3 expression was upregulated (in *Etv6*-deficient embryos or by overexpressing *foxo3* mRNA) and this repression was abolished when the Foxo3 consensus binding site was deleted. This confirms the proposed GRN.

In conclusion, Foxo3 binds the conserved Foxo3 binding site in the *vegfa* promoter and represses its transcriptional activity.

Is the control of *vegfa* expression by this ETV6 GRN unique to the somites at stage 22 or is this also seen elsewhere during development?

We have previously shown through loss-of-function studies that, between stages 20 and 22, *vegfa* expression in the somites is dependent on *Etv6* (Ciau-Uitz *et al.*, *Development*, 2013). During this period of development, *vegfa* is also expressed in the head and the hypochord, but expression in these tissues is not *Etv6* dependent. Later, by stage 24, *vegfa* expression can be detected in the somites of *Etv6*-deficient embryos. This indicates that *vegfa* expression is no longer under *Etv6* control and, therefore, *Etv6* regulates *vegfa* expression in the somites in a very narrowly defined developmental window, i.e. at stages 20 to 22.

We have also shown that *Etv6* controls the expression of *vegfa* in definitive hemangioblasts located in the lateral plate mesoderm, downstream of *VegfA* signaling from the somites. Here, *VegfA* secreted by the somites binds its receptor *Kdr* on definitive hemangioblasts. This activates *Etv6* expression and *Etv6*, in turn, activates *vegfa* expression. Whether the same *Etv6* GRN operates in the somites and hemangioblasts requires investigation. However, *Klf4* is not expressed in hemangioblasts, suggesting that distinct GRNs operate in these tissues.

To our knowledge, there is no evidence that *Etv6* controls *vegfa* expression in other tissues during development.

Minor comments:

Line 170: “Surprisingly, the majority of Etv6 target genes (303/482, 63%) were downregulated in the somites of etv6-deficient embryos”. Given that “only” 63% of the genes are downregulated, it might be more appropriate to write “a large fraction” or “around two third” rather than “the majority”.

We thank the Reviewer for this question as it has made us realise that we mistakenly reported the number of Etv6 target genes from an early and incomplete list. The correct number of genes on which our analysis was based is 540, which corresponds to 545 peaks in the TSS regions. Nevertheless, the proportion of genes downregulated in Etv6-deficient embryos reported in this list (339/540, 63%) is similar to what we erroneously indicated in the text (303/482, 63%). Importantly, the number of TFs reported as Etv6 direct targets (Figure 2f) was right. This correction, therefore, does not alter our results. The manuscript has been updated according to this data (pages 9, 18/19, Figure 2e, **Supplementary Table 6**). We deeply apologise for this mistake.

The text has also been modified according to the Reviewer’s suggestion (page 9).

Line 171-172: “strongly indicating that Etv6, typically considered as a transcriptional repressor, acts mainly as an activator of gene”. Rather than “mainly”, it might be more appropriate to use the word “also”.

The text has been modified according to the Reviewer’s suggestion (page 9).

--

Reviewer #3 (Remarks to the Author):

Here, Lei Li and coworkers examine the role of the transcription factor Etv6 in regulation of Vegfa expression during *Xenopus laevis* development. Main findings are that Etv6 suppresses expression of the negative regulatory Foxo3 while it enhances expression of the positive regulatory Klf4. Moreover, Klf4 recruits Etv6 to the Vegfa promoter. The fact that TFs act alone or in complex with other factors to both enhance and repress gene regulation is of course well established already also for Etv6 and was suggested by Ciau-Uitz et al., to explain the mode of action of Etv6 in regulation of Vegfa expression in their 2010 Dev Cell paper. Moreover, reports on Foxo3 as a repressor and Klf4 as an inducer of Vega transcription have been published. The role of Etv6 in repression and activation of Foxo3 and Klf4, respectively, presented here, is new. Also, not as much is known about gene regulation in *X. laevis*, which is the preferred model of the Porcher laboratory, as in other genetic models. The study is well presented. Overall, however, comprehensive aspects of Vegfa gene regulation and Etv6/TF mode of action are missing.

1. To map Etv6 binding sites in the *X. laevis* genome, anti-Etv6 antibodies were raised for DNA immunoprecipitation followed by sequencing. The authors list a few genes showing the highest enrichment of Etv6 in their transcription start site (rimklb and Xelaev18005411m). Vegfa is not in this list and the two genes are just mentioned which is not very meaningful.

We apologise for the lack of clarity in the choice of the examples provided in Figure 1c. As we raised the anti-*Xenopus* Etv6 antibodies used in this study, we felt it was important to present ChIP-seq profiles simply to illustrate the quality of the ChIP-seq data (and, thus, of the antibodies) and to provide examples of positive and negative ChIP peak calling. For the examples of Etv6 binding, we chose to show the two genes harboring the strongest Etv6 peaks. This has been clarified in the text (page 6). Only after we performed RNA-seq and

functional analysis did we focus on genes directly transcriptionally regulated by Etv6 and relevant for *vegfa* regulation (*foxo3* and *klf4*).

Regarding *vegfa*, we do present Etv6 ChIP-seq on *vegfa* in Figure 5. The reason why we did not focus on *vegfa* earlier in the study is because, prior to this report, Etv6 had only been shown to be a transcriptional repressor. As *vegfa* expression in the somites is positively regulated by Etv6 (our previous report, Ciau-Uitz *et al.*, *Dev Cell*, 2010), our original hypothesis was that Etv6 would not directly bind *vegfa*'s regulatory sequences but, instead, would down-regulate the expression of a transcriptional repressor which, otherwise, would bind and repress *vegfa* regulatory sequences. Therefore, we first studied the transcriptional regulation of *foxo3* (and of *klf4*, as we discovered that Etv6 also acts as an activator) and their role in *vegfa* expression. Only thereafter did we examine the binding and activity of Etv6 observed on the *vegfa* promoter.

The authors go on to conclude that Etv6 peaks in the TSS analysis do not contain ETS family binding motifs but show motifs compatible with binding of other TFs and that Etv6 exerts its gene regulatory effect by complex formation with a range TFs. Here, the authors describe only the TSS regions which make up about 25% of the Etv6 peaks in the ChipSeq. Why not analyze the remaining Etv6 peaks as well? Are findings only relevant to the TSS region?

We chose to focus on TSS regions as a means to unambiguously link a peak to the gene it might regulate.

To follow the Reviewer's suggestion and investigate whether Etv6 might be recruited to DNA genome-wide through interaction with other TFs rather than direct DNA binding, we performed *de novo* motif analysis on all the genomic sequences targeted by Etv6 (new Supplementary Table 3). As for the TSSs, this analysis shows that the most overrepresented binding motif corresponds to the Klf/Sp family of binding factors (28.29%, $P=1e-1060$) and that ETS binding motifs are present in only a small fraction of all Etv6 ChIP peaks (3.31%, $P=1e-13$). Therefore, our findings on TSS peaks are relevant genome-wide: Etv6 recruitment to regulatory sequences is largely dependent on other DNA-binding TFs such as members of the Klf/Sp family. This is now indicated in the Results section page 7.

2. To obtain a more comprehensive analysis of Vegfa gene regulation, it would have been interesting to see a map all known TF motifs in the Vegfa gene.

As suggested, we show all potential binding sites in the *vegfa* promoter, after analysis through the Jaspar database (see Supplementary Table 8). As expected, a large number of potential TF binding sites was predicted in the *vegfa* promoter region (the analysis was performed on a genomic fragment of ~2kb upstream of the TSS, that contains a ~700 bp region for which we have no sequence information). Although of prime interest to fully understand how *vegfa* is regulated, studying the functional relevance of these binding sites in the regulation of *vegfa* is beyond the scope of this study.

Some of these TFs would be Etv6 regulated, as mentioned in the first part of the discussion, others would not. The limitation of this approach would be that distant enhancers would not be detected. However, now, the analysis is focused rather narrowly to Foxo3 and Klf4 based on that they bind to the Vegfa TSS and that their expression is regulated by Etv6. Would TF genes need to be repressed/activated by Etv6 for it to engage in TF complexes regulating Vegfa expression?

We agree with the Reviewer that our approach does not allow the identification of all the regulators of *vegfa* expression. This was, however, not our aim. The starting point of this study was that Etv6, an established transcriptional repressor, indirectly regulates *vegfa* expression in the somites at stage 22 (Ciau-Uitz *et al.*, *Dev cell*, 2010). We therefore sought to identify Etv6 target genes that could control *vegfa* expression in this

tissue at this stage of development. As mentioned in the response to point 1, we deliberately chose to focus on genes bound by Etv6 at their TSSs as we did not want to rely on the nearest gene approach to connect Etv6 peaks with putative target genes. This has limited the number of target genes identified but, nevertheless, has allowed the characterization of two important regulators of *vegfa* expression, Foxo3 and Klf4, in the somites during blood development. These two TFs were chosen as they were known direct regulators of *VegfA* expression in other tissues (see page 10). The GRN we describe, based on the activity of these 2 regulators, shows the complexity of *vegfa* regulation (coherent feed-forward loop and double negative gate), reflecting the requirement for a tight control of *vegfa* expression. This GRN will certainly grow in complexity as more regulatory pathways are investigated. We anticipate that some of the additional TFs regulating *vegfa* expression will not be controlled by Etv6.

3. The authors emphasize that Klf4 recruits Etv6 to the Vegfa promoter as enrichment of Etv6 detected by ChipSeq was decreased upon deletion of Klf4 binding motifs in the Vegfa promoter. Would not Foxo3 also recruit Etv6 to the Vegfa promoter?

This is a very interesting hypothesis. However, three lines of evidence indicate that, under normal conditions, Foxo3 is unlikely to recruit Etv6 to the *vegfa* promoter in the somites at stage 22 of development:

(1) The conserved Foxo3 consensus binding site identified in the *vegfa* promoter region does not overlap with the Etv6 ChIP peak but localizes over 600bp upstream of the TSS (this is now clearly shown in Supplementary Table 8 and Supplementary Figure 7).

(2) Etv6 occupancy at the Foxo3 binding site is not detected by ChIP-seq.

(3) In wild-type embryos, *foxo3* expression is barely detectable by WISH in the somites at stage 22 of development (Figure 3c). Conversely, in Etv6-deficient embryos (*etv6* MO), *foxo3* expression is increased but Etv6 is not present. Therefore, Foxo3 and Etv6 are not co-expressed in either wild-type or Etv6-deficient somitic cells and, in combination with the evidence presented above, highly unlikely to interact with each other.

Minor

4. The comment in the end of the abstract on the potential usefulness of the presented data in new strategies for treatment of cancer, should be removed. Clearly, this study does not have any bearing on tumor therapy and to add such statements as is often done, just takes credit away.

We thank the Reviewer for this comment. The sentence has been removed.

5. In the intro, second para, it says that knockout of VegfA receptor (Vegfr1,2 or 3) genes results in early lethality – VEGFR3 is not a receptor for VEGFA, so please adjust.

Some evidence indicates that Vegfr2-Vegfr3 heterodimers can actually transduce VegfA signaling (Nilsson *et al.*, *EMBO J*, 2010; Domigan *et al.*, *ATVB*, 2015). Therefore, Vegfr3 can function as a VegfA receptor. However, to avoid confusion, this statement has been modified as suggested by the Reviewer.

6. In the results part under “Establishing the transcriptome regulated by Etv6 in the somites”, please motivate why stage 22 embryos were used for the general readership.

The rationale for using stage 22 embryos is clearly stated in the last paragraph of the introduction. Nevertheless, this has been reiterated as suggested by the Reviewer (page 7).

--

Reviewer #4 (Remarks to the Author):

The authors performed Etv6 ChIP-seq using somites of the stage 22 frog embryos to understand the mechanisms by which Etv6 regulates Vegfa expression. While Etv6 is believed to repress its target genes, they found Etv6 to both activate and repress its target genes. While repressed genes have an ETS motif in their promoter regions, activated genes don't have an ETS site. Importantly, they identified potential activator as well as repressor of Vegfa expression. For example, Klf4, a presumed activator of Vegfa is activated by Etv6, and the authors propose a model that Etv6 activates Klf4, which in turn recruits Etv6 to the Vegfa promoter region, and activates Vegfa expression. While this study is the first to perform Etv6 ChIP-seq and suggest Etv6 can function both as an activator and repressor of the target genes, data sometimes is indirect and incomplete in supporting their argument. Particularly, the study falls short of providing mechanistic insights of how Etv6 can function both as an activator and repressor at the same time. Vegfa expression in somites does not necessarily reflect direct linear causal relationship of the various manipulations.

Overall, target gene validation is not optimal. Peak enrichment on the target site is not sufficient to establish that the identified genes are indeed direct target genes. All potential target genes should be validated by taking the potential binding regions and performing luciferase reporter assays, at least.

To clarify our approach, we have used a combination of both ChIP-seq and RNA-seq experiments as a powerful strategy to identify potential direct target genes *in vivo*, in the tissue and developmental stage of interest: these genes are defined based on both Etv6 binding in their promoter and the dysregulation of their expression in absence of Etv6. However, we fully agree with the Reviewer that functional experiments need then to be carried out to establish that the binding regions are indeed directly involved in the regulation of gene expression. In the first version of this manuscript, we specifically tested the function of the Etv6 peak at the *vegfa* promoter by deleting the Klf4 binding sites through Talen-mediated mutagenesis (Figure 5f-h). This provided strong *in vivo* evidence for the regulatory activity of this sequence.

As suggested by the Reviewer, we have now performed luciferase assays in embryos to test the function of the Etv6-bound sequences associated with the promoters of the other two direct targets we focused on in this study: *klf4* and *foxo3*. Additionally, as asked by Reviewer 2, we have performed luciferase assays using the DNA fragment of the *vegfa* promoter containing the Foxo3 consensus binding site. Importantly, the vectors were injected in embryos on their own or co-injected with *etv6* MO (and *foxo3* mRNA in the case of the Foxo3 binding sequences) to perform analyses on wild-type (WT), Etv6-deficient or *foxo3* overexpressing backgrounds, respectively. All the DNA fragments tested showed the predicted transcriptional activities, thus confirming our initial conclusions and strengthening the relevance of the Etv6 GRN regulating *vegfa* expression in the somites. These experiments are now presented in Supplementary Figures 3 and 7 and referred to in the text (pages 11-14).

The mechanisms by which Etv6 functions as an activator or repressor is not clear. For example, the idea they like the readers to believe is that Etv6 represses its target genes through an ETS site. I would like to see that the authors take several repressed genes and perform luciferase assay to show that indeed these TSS regions

of the repressed genes are responsible for Etv6 repression. Also, I would like to see the mutational analysis of the ETS site(s) of the target genes, and when mutated, the reporter expression in turn goes up. As for the activated genes, I also want to see the similar types of data.

We thank the Reviewer for the opportunity to clarify a few points.

First of all, we would like to apologise as we have realized, whilst addressing the Reviewer's comment, that the genes reported as up- or down-regulated in the *de novo* motif analysis had been interchanged. The genes were, however, correctly categorised in the RNA-seq analysis. As a consequence, the motifs associated to the genes normally activated were in fact associated to genes normally repressed, and *vice versa*. This has not changed our main findings and conclusion as the same consensus binding sites are observed in the two lists of genes (Klf/Sp, Nfy and Creb), except for Hox motifs, now associated to genes normally repressed instead of normally activated (but Hox motifs were only mentioned and not further studied in the original manuscript). Moreover, as discussed in the next paragraphs, we no longer suggest that ETS binding motifs (originally associated to repressed genes) could help distinguish Etv6 transcriptional activities. We have corrected this data throughout the manuscript (pages 10, 19, and Supplementary Table 7).

Regarding the transcriptional activities of Etv6, we totally agree that the mechanisms by which Etv6 activates or represses gene expression are not clear. Indeed, our study unexpectedly unveiled the fact that Etv6, considered to be a transcriptional repressor, can directly activate gene expression. Understanding how Etv6 can perform this dual function is clearly now an important biological and mechanistic question. We have, however, not focused on elucidating this mechanism in our manuscript. We acknowledge that we have generated confusion about the enrichment and potential function of the ETS-binding motifs in the transcriptional activity of Etv6, and apologise for suggesting that presence or absence of ETS binding motifs at Etv6 peaks may allow the discrimination between activated and repressed genes.

To clarify our data, *de novo* motif analysis shows that only a small fraction (3.31%) of all Etv6 genomic targets are enriched with ETS-binding motifs (TACTTCTT, Supplementary Table 3, text page 7). This trend is also applicable to peaks associated with differentially regulated genes as only 9.94% of those associated with genes normally activated by Etv6 are enriched in this motif (CTTCCGCCCTTT and GGCCGGCAGTGT, see Supplementary Table 7 and text page 10). Therefore, the impact of the ETS-binding motif in the dual regulatory function of Etv6 may only be relevant (if relevant at all) in a small fraction of the genes it regulates, and does not therefore underlie a general mechanism. We have removed the sentence in the text suggesting that Etv6 binding could allow to distinguish Etv6 transcriptional activities (page 10). Clearly, this was an over-interpretation of our data.

Elucidating the mechanisms by which Etv6 activates and represses gene expression will require considerable effort beyond the scope of this report. At the moment, we can only suggest potential mechanisms. For example, as KLF4 has been shown to both activate and repress gene expression in a context-dependent manner (Ghaleb *et al.*, Gene, 2017), one hypothesis is that Etv6 activating and repressive activities may be conferred by Klf4. As the mechanisms underlying KLF4 dual transcriptional activity are currently not known, this will require investigation. Separately, as Klf/Sp motifs are enriched in around 60% of the peaks associated with both genes activated and repressed by Etv6, another possible mechanism would be that distinct Klf/Sp proteins interact with Etv6 to drive transcriptional activation or repression. These hypotheses actually form the basis of a grant application currently under assessment that will allow us to take this project to a more mechanistic level.

The authors propose that Klf4 recruits Etv6 to activate Vegfa expression. I would like to see the direct interaction between Klf4 and Etv6.

As requested by the Reviewer (and also by Reviewer 2), we have performed co-IP experiments to test whether Klf4 directly interacts with Etv6 or requires auxiliary proteins to recruit Etv6 to the *vegfa* promoter.

We first performed an *in vivo* co-IP experiment on stage 22 somitic protein extracts. As immunoprecipitation on endogenous Etv6 protein did not reveal interaction with Klf4 (likely due to the low level of expression of this TF), we next overexpressed exogenous Etv6 protein in developing embryos and then assayed its capacity to co-immunoprecipitate endogenous Klf4 protein. This experiment indicated that Klf4 co-purifies with Etv6 (Figure 6a, and text page 16/17).

To test if this interaction occurs directly, Etv6 and Klf4 proteins were then produced by *in vitro* translation using the rabbit reticulocyte lysate system and then used in co-IP assays. In these conditions, these two proteins did not co-immunoprecipitate (Figure 6b, and text page 17). In contrast, as a positive control, two proteins known to directly interact, namely Lmo2 and Ldb1 (Lahlil *et al.*, *MCB*, 2004), did co-IP in the assay (Figure 6c). This result indicates that, *in vitro*, there is no direct physical interaction between Etv6 and Klf4.

Taken together, these experiments show that Klf4 and Etv6 are part of multiprotein complexes and suggest that Klf4 recruits Etv6 to the *vegfa* promoter through auxiliary proteins.

The co-IP protocols have been added to the Methods section (page 30/31).

Can the authors show that Klf4 can rescue Etv6 MO effect on Vegfa, Tal1, and Runx1 expression?

To answer this question, exogenous *klf4* was overexpressed in Etv6-deficient embryos and expression of *vegfa* in the somites, *tal1* in the lateral plate mesoderm and *runx1* in the dorsal aorta was assessed. *Klf4* overexpression did not rescue the expression of these genes in Etv6-deficient embryos (Supplementary Figure 12 and text page 17). This strongly supports the notion that Etv6-driven *vegfa* expression in the somites (a pre-requisite for hematopoietic development and, thus, *tal1* and *runx1* expression - Ciau-Uitz *et al.*, *Development*, 2010) does not solely rely on activation of *klf4* expression. It confirms that Etv6 has other regulatory functions, such as repressing *foxo3* expression and binding the *vegfa* promoter. Together with the fact that exogenous *vegfa* can rescue *tal1* and *runx1* expression in Etv6 morphants (Ciau-Uitz *et al.*, *Dev Cell*, 2010), this strongly indicates that Klf4's primary role in hematopoietic stem cell programming is to activate *vegfa* expression in the somites through the recruitment of Etv6 to the *vegfa* promoter.

I cannot seem to find the data showing that Etv6 expression is not affected by either Foxo3a or Klf4 MO treatment, which is critical.

Data showing that *etv6* expression is not affected by *klf4* MO treatment was shown in old Supplementary Figure 7 (now Supplementary Figure 10).

The expression of *etv6* in Foxo3 MO was not analyzed because *foxo3* expression is barely detectable in wild type embryos by WISH and, as expected, the MO has no overt phenotype on *vegfa* expression. In contrast, *foxo3* expression is detected in Etv6-deficient somites. Therefore, to investigate the effects of Foxo3 on *etv6* expression, exogenous *foxo3* was injected and *etv6* expression quantified by WISH and RT-qPCR. *Foxo3*

overexpression did not affect *etv6* expression in the somites. This data is presented in new Supplementary Figure 4 and text page 12.

Reviewers' Comments:

Reviewer #1:

Remarks to the Author:

The revisions made by the authors address all the concerns that I raised. In addition, they have performed several more experiments that provide more information about the interactions among the multiple transcription factors regulating vegfa. Thus an already strong paper is even more thorough and informative.

Ross Hardison

Reviewer #2:

Remarks to the Author:

The authors have addressed all my comments.

Reviewer #3:

Remarks to the Author:

The authors have responded adequately to the criticisms. I have no further comments.